# Model Sensitivity Aware Continual Learning

**Zhenyi Wang and Heng Huang**
Department of Computer Science, Institute of Health Computing
University of Maryland College Park
College Park, MD, 20742
zwang@umd.edu;heng@umd.edu

## Abstract

Continual learning (CL) aims to adapt to non-stationary data distributions while retaining previously acquired knowledge. However, CL models typically face a trade-off between preserving old task knowledge and excelling in new task performance. Existing approaches often sacrifice one for the other. To overcome this limitation, orthogonal to existing approaches, we propose a novel perspective that views the CL model ability in preserving old knowledge and performing well in new task as a matter of model sensitivity to parameter updates. *Excessive* parameter sensitivity can lead to two drawbacks: (1) significant forgetting of previous knowledge; and (2) overfitting to new tasks. To reduce parameter sensitivity, we optimize the model's performance based on the parameter distribution, which achieves the worst-case CL performance within a distribution neighborhood. This innovative learning paradigm offers dual benefits: (1) reduced forgetting of old knowledge by mitigating drastic changes in model predictions under small parameter updates; and (2) enhanced new task performance by preventing overfitting to new tasks. Consequently, our method achieves superior ability in retaining old knowledge and achieving excellent new task performance simultaneously. Importantly, our approach is compatible with existing CL methodologies, allowing seamless integration while delivering significant improvements in effectiveness, efficiency, and versatility with both theoretical and empirical supports.

## 1 Introduction

Continual learning (CL) embodies a dynamic approach aimed at adapting to non-stationary data distributions that evolve over time. However, in pursuit of this goal, CL encounters a significant challenge: the trade-off between preserving previously acquired knowledge and effectively learning new knowledge. As the model assimilates new information, it often swiftly erases previously learned knowledge, resulting in catastrophic forgetting (CF) on past tasks [44, 54]. Effectively addressing CF during CL is essential to preserve previously acquired information. On the other hand, effectively learning new information is equally crucial for CL models to adapt to new tasks and environments.

Existing approaches in CL often face a dilemma: they either prioritize preserving old knowledge or excelling in new task performance, often at the expense of the other. When a CL model prioritizes preserving old knowledge, it focuses on retaining information from previous tasks while minimizing interference or forgetting. However, excessive emphasis on old knowledge can limit the model's ability to adapt to new tasks. Conversely, when a model prioritizes new task performance, it aims to quickly adapt to new tasks or data distributions. Yet, this emphasis on new tasks can potentially degrade performance on previously learned tasks.

To overcome the aforementioned limitations, orthogonal to existing approaches, we introduce the concept of model sensitivity and approach the challenge of balancing old knowledge retention and new task performance in CL from the perspective of model parameter sensitivity. When a CL

38th Conference on Neural Information Processing Systems (NeurIPS 2024).

model exhibits high sensitivity to parameter changes, it leads to two significant issues: (1) *Increased Forgetting*: Excessive sensitivity in model parameters can cause abrupt and substantial changes in model predictions with minor parameter adjustments during CL. This phenomenon results in significant forgetting of previous tasks. (2) *Diminished New Task Performance*: High sensitivity in model parameters can also result in severe overfitting on new tasks. Overfitting occurs when a model memorizes the training data instead of generalizing patterns that can be applied to unseen data. High parameter sensitivity means that even minor alterations in the training data can induce substantial modifications in the learned model. This renders the model excessively tailored to the training data and reduces its adaptability to new, unseen data, consequently leading to suboptimal performance on new tasks.

To reduce the CL model parameter sensitivity under model updates, we aim to ensure that even minor alterations in model parameters do not substantially impair CL model performance. This is accomplished by optimizing the model's performance based on the worst-case scenario of parameter distributions within a distribution neighborhood. However, finding the optimal worst-case CL model parameter distribution is challenging since the space of all possible distributions within the neighborhood is an infinite-dimensional space [32]. To efficiently solve this problem, we parameterize the optimal worst-case CL model parameter distribution as Gaussian distribution. We propose a natural-gradient descent (NGD) method to efficiently inference the mean and covariance of the Gaussian distribution since NGD incorporates the information geometry of the parameter space by adapting the step size based on the curvature of the cost function. This adaptive approach leads to faster convergence compared to conventional gradient descent methods, particularly in high-dimensional spaces where the curvature exhibits notable variations. This is especially beneficial for CL models. However, calculating the natural gradient is computationally expensive due to the explicit calculation of Fisher information matrix (FIM). We thus update the worst-case CL parameters in the expectation parameter space, rather than the traditional natural parameter space, of the Gaussian distribution, thereby eliminating the need for explicit calculation of the FIM.

Our method offers dual benefits: (1) *Reduced Forgetting*: By mitigating parameter sensitivity and avoiding drastic changes in model prediction, our approach effectively reduces the loss of previously learned task knowledge. (2) *Improved New Task Performance*: Through decreased parameter sensitivity, the model becomes less susceptible to overfitting on new task training data. This reduced vulnerability to minor fluctuations fosters the learning of more generalized patterns rather than memorizing specific examples. As a result, the model demonstrates enhanced generalization capabilities on new tasks. Therefore, our method simultaneously achieves superior performance in retaining previously learned knowledge and excelling in new task performance.

We provide a thorough theoretical analysis for our method. Firstly, the theory illustrates that our approach implicitly reduces the variance of loss against different parameter variations, thereby indicating reduced model parameter sensitivity. Secondly, our method tightens the generalization bound of CL models, suggesting enhanced generalization. Furthermore, our extensive experiments across multiple datasets, compared to various state-of-the-art (SOTA) baseline methods, reveal substantial enhancements in overall performance across all learned tasks, backward transfer, and new task test accuracy. These results indicate significantly enhanced CL model ability in preserving old knowledge and achieving better performance on new task with our method. Additionally, our proposed approach seamlessly integrates with existing CL methodologies, functioning as a versatile plug-in. This demonstrates the effectiveness, efficiency, and versatility of our method.

Our contributions can be summarized as follows:

- We tackle the challenge of both retaining old task knowledge and excelling in new task in CL from a novel perspective by mitigating model parameter sensitivity.

- We introduce a novel CL approach aimed at reducing model parameter sensitivity by optimizing CL model performance under the worst-case parameter distribution within a distribution neighborhood. Additionally, we propose an efficient learning algorithm to identify the worst-case parameter distribution.

- We provide comprehensive theoretical analyses that substantiate our method's ability to decrease model parameter sensitivity and improve model generalization.

- Extensive experiments conducted across multiple datasets demonstrate the efficacy and versatility of our proposed method.

## 2 Related Works

CL aims to learn non-stationary data distributions without forgetting previously learned knowledge. The CL scenarios can be further categorized into three scenarios: task-incremental learning (Task-IL), domain-incremental learning (Domain-IL) and class-incremental learning (Class-IL) [67]. Task-IL and Class-IL are most representative scenarios in CL, we thus focus on these two scenarios. Existing approaches for CL can be categorized into five classes: (1) *regularization-based* methods incorporate regularization terms either in model weights or outputs into the loss function to mitigate catastrophic forgetting when learning new tasks, including [28, 62, 84, 55, 11, 1, 22, 10, 39]; (2) *memory replay-based* methods address the challenge of catastrophic forgetting by explicitly storing and replaying a subset of past experiences (samples from previous tasks) while learning new tasks, including [40, 57, 15, 7, 51, 68, 3, 8, 75, 4, 74, 76, 61, 78, 77, 83, 36, 73, 72]; (3) *gradient-projection-based* methods aim to mitigate catastrophic forgetting by projecting gradient updates onto subspaces that minimize interference with previously learned tasks, including [13, 17, 60, 71, 38, 52, 82]; (4) *architecture-based* methods involve dynamically adapting and modifying the neural network architecture to accommodate new tasks while preserving performance on previously learned tasks, including [41, 63, 34, 23]; (5) *Bayesian-based* methods leverage principles from Bayesian inference to manage the uncertainty and learning of new tasks while preserving knowledge from previous tasks, including [48, 58, 30, 25, 21, 49, 66, 59].

In contrast to existing methods, which often necessitate a trade-off between retaining old knowledge and learning new knowledge, sacrificing one for the other, our approach takes a different path. It sets itself apart from these existing methods by offering an orthogonal solution that preserves old task knowledge while simultaneously enhancing new task performance. This novel perspective is achieved by reducing parameter sensitivity.

Connection with existing flat-minima/SWAD approaches: (1) Connection and difference with sharpness-aware minimization (SAM) [18, 27, 45] related approach: Our method is fundamentally different from SAM-based CL in two aspects. (i) *Deterministic vs. Probabilistic Approach*: SAM uses a fixed deterministic neighborhood, which can be restrictive in practice since updates are constrained within a fixed ball. In contrast, our method employs a probabilistic distributional approach, offering two distinct advantages: (a) The distributional neighborhood is more flexible and covers a broader range of parameter variations by sampling from a neighborhood distribution, and (b) Stochastic Gradient Descent (SGD) introduces noise during CL. Our distributional approach accounts for this noise, making it a more realistic model in practice and providing stronger guarantees against parameter sensitivity. (ii) *Uniform vs. Parameter-specific sensitivity without explicit calculation of FIM*: SAM uniformly updates all parameters, overlooking the varying importance and sensitivity of each parameter in the context of CL. Our method, on the other hand, considers these differences and treats parameters uniquely through the natural gradient without needing to explicitly calculating the FIM. This distinction is crucial for CL, as each parameter has different sensitivity to forgetting—a factor that SAM does not address. (2) Connection and difference with model averaging flatness seeking approach: SWA [24] and SWAD [9], which aim to achieve flatter minima by averaging multiple models during training. However, these approaches are memory-intensive and inefficient for CL, as they require storing multiple sets of model parameters, which compromises memory efficiency.

## 3 Method

In this section, we first present the preliminary in section 3.1 and then present the model sensitivity aware continual learning in section 3.2.

### 3.1 Preliminary

**Continual Learning Setup** The standard CL problem involves learning a sequence of $T$ tasks, represented as $\mathcal{D}^{tr} = \{\mathcal{D}_1^{tr}, \mathcal{D}_2^{tr}, \cdots, \mathcal{D}_T^{tr}\}$. The training dataset $\mathcal{D}_k^{tr}$ for the $k^{th}$ task contains a collection of triplets: $(\boldsymbol{x}_i^k, y_i^k)_{i=1}^{n_k}$, where $\boldsymbol{x}_i^k$ denotes the $i^{th}$ data example specific to task $k$, $y_i^k$ represents the associated data label for $\boldsymbol{x}_i^k$. The primary objective is to train a neural network function, parameterized by $\boldsymbol{\theta}$, denoted as $g_{\boldsymbol{\theta}}(\boldsymbol{x})$. The goal is to achieve good performance on the test datasets from all the learned tasks, represented as $\mathcal{D}^{te} = \{\mathcal{D}_1^{te}, \mathcal{D}_2^{te}, \cdots, \mathcal{D}_T^{te}\}$, while ensuring that knowledge

acquired from previous tasks is not forgotten. The CL loss function is defined as the following:

$$\mathcal{L}^{CL}(\boldsymbol{\theta}) := \mathcal{L}_{CE}(\boldsymbol{x}, y; \boldsymbol{\theta}) + \zeta \mathcal{L}_f(\boldsymbol{\theta}) \tag{1}$$

where $\mathcal{L}_{CE}(\boldsymbol{x}, y; \boldsymbol{\theta})$ is the current task cross-entropy loss function. $\mathcal{L}_f(\boldsymbol{\theta})$ is the forgetting-mitigation loss, e.g., memory-replay, weight-regularization and gradient-projection loss, etc. $\zeta$ is a constant that balances the weight between the loss of the new task and the loss of the previous tasks.

**Exponential Family of Distributions**    The exponential family distribution [70] is defined as:

$$P_{\boldsymbol{\phi}}(\boldsymbol{\theta}) := h(\boldsymbol{\theta})exp(\langle \boldsymbol{\phi}, \boldsymbol{\Omega}(\boldsymbol{\theta})\rangle - Z(\boldsymbol{\phi})) \tag{2}$$

Where := denotes a definition. In existing literature [70], $\boldsymbol{\phi}$ are called the natural parameters for defining the distribution, $P_{\boldsymbol{\phi}}(\boldsymbol{\theta})$. $h(\boldsymbol{\theta})$ is the base measure, $\boldsymbol{\Omega}(\boldsymbol{\theta})$ is the sufficient statistic, $Z(\boldsymbol{\phi}) := \log \int h(\boldsymbol{\theta})exp(\langle \boldsymbol{\phi}, \boldsymbol{\Omega}(\boldsymbol{\theta})\rangle)d\boldsymbol{\theta}$ is the log-partition function, $\langle, \rangle$ denotes the dot product between two vectors. We denote the expectation parameters as $\boldsymbol{\lambda} := \mathbb{E}_{P_{\boldsymbol{\phi}}(\boldsymbol{\theta})}\boldsymbol{\Omega}(\boldsymbol{\theta})$. We can write multivariate Gaussian distribution as canonical form of exponential family as:

$$f(\boldsymbol{\theta}; \boldsymbol{\mu}, \boldsymbol{\Sigma}) := \frac{1}{(2\pi)^{\frac{d}{2}} det(\boldsymbol{\Sigma})^{\frac{1}{2}}} exp\{-\frac{1}{2}(\boldsymbol{\theta} - \boldsymbol{\mu})^T \boldsymbol{\Sigma}^{-1}(\boldsymbol{\theta} - \boldsymbol{\mu})\} \tag{3}$$

$$= exp\{\boldsymbol{\theta}^T \boldsymbol{\Sigma}^{-1}\boldsymbol{\mu} - \frac{1}{2}\boldsymbol{\theta}^T \boldsymbol{\Sigma}^{-1}\boldsymbol{\theta} - \frac{1}{2}[d\log 2\pi + \log|\boldsymbol{\Sigma}| + \boldsymbol{\mu}^T \boldsymbol{\Sigma}^{-1}\boldsymbol{\mu}]\} \tag{4}$$

Therefore, the correspondence between $f(\boldsymbol{\theta}; \boldsymbol{\mu}, \boldsymbol{\Sigma})$ and exponential family distribution in Eq.(2) can be expressed as the following:

$$\boldsymbol{\phi} := (\boldsymbol{\Sigma}^{-1}\boldsymbol{\mu}, -\frac{1}{2}\boldsymbol{\Sigma}^{-1}), \quad \boldsymbol{\Omega}(\boldsymbol{\theta}) := (\boldsymbol{\theta}, \boldsymbol{\theta}\boldsymbol{\theta}^T) \tag{5}$$

$$\boldsymbol{\lambda}^1 := \mathbb{E}_{f(\boldsymbol{\theta};\boldsymbol{\mu},\boldsymbol{\Sigma})}\boldsymbol{\theta} = \boldsymbol{\mu}, \quad \boldsymbol{\lambda}^2 := \mathbb{E}_{f(\boldsymbol{\theta};\boldsymbol{\mu},\boldsymbol{\Sigma})}\boldsymbol{\theta}\boldsymbol{\theta}^T = \boldsymbol{\mu}\boldsymbol{\mu}^T + \boldsymbol{\Sigma} \tag{6}$$

Derivations details of Eq.(6) can be found in Appendix B.1. In the following section, we use exponential family distributions to parameterize the worst-case of CL model parameter distribution since this enables us to efficiently calculate the natural gradient in the expectation parameter space $\boldsymbol{\lambda}$ without needing to explicitly calculate the Fisher information matrix (FIM) in natural parameter space $\boldsymbol{\phi}$.

## 3.2    Model Sensitivity Aware Continual Learning

**Learning Objective**    Specifically, we propose the following CL learning objective to reduce the CL parameter sensitivity under model parameter updates:

$$\min_{\boldsymbol{\mu}} \max_{\mathbb{U} \in \mathcal{U}} \mathbb{E}_{\boldsymbol{\theta} \sim \mathbb{U}(\boldsymbol{\theta})} \mathcal{L}^{CL}(\boldsymbol{\theta}) \tag{7}$$

$$\text{s.t. } \mathcal{U} = \{\mathbb{U} : D_{\mathrm{KL}}(\mathbb{U}, \mathbb{V}) \leq \epsilon\}$$

where $\mathcal{U}$ denotes the uncertainty set. $D_{\mathrm{KL}}(\mathbb{U}, \mathbb{V})$ denotes the KL divergence between the current CL model parameter distribution $\mathbb{V}$ and the neighbour CL model parameter distribution $\mathbb{U}$. $\epsilon$ is a small constant. $\max_{\mathbb{U} \in \mathcal{U}} \mathbb{E}_{\boldsymbol{\theta} \sim \mathbb{U}(\boldsymbol{\theta})} \mathcal{L}^{CL}(\boldsymbol{\theta})$ aims to find the worst-case CL model parameter distribution within a neighbourhood. We choose probabilistic distributional neighbourhood due to two-fold reasons: (1) the distributional neighbourhood covers more flexible parameter space; and (2) widely used SGD method incurs update noise during CL, thereby distributional neighbourhood provides stronger guarantee against parameter sensitivity. It is important to note that the outer minimization is performed with respect to $\boldsymbol{\mu}$, the expectation of $\boldsymbol{\theta}$, since during inference, only $\boldsymbol{\mu}$ is used as the model parameter for predictions.

**Objective for Learning the Worst-Case CL Parameter Distribution**    We convert the constrained inner maximization optimization in Eq. (7) into the following unconstrained optimization to find the worst-case CL model parameter distribution.

$$\arg\min_{\mathbb{U}}[H(\mathbb{U}) := -\mathbb{E}_{\boldsymbol{\theta} \sim \mathbb{U}(\boldsymbol{\theta})} \mathcal{L}^{CL}(\boldsymbol{\theta}) + \alpha D_{\mathrm{KL}}(\mathbb{U}, \mathbb{V})] \tag{8}$$

where $\alpha > 0$ is a constant. However, solving Eq. (8) is intractable since the optimization target is in an infinite-dimensional function space [32]. For computation efficiency, we set the current CL

model parameter distribution as $\mathbb{V}(\boldsymbol{\theta}) = \mathcal{N}(\boldsymbol{\theta}|\boldsymbol{\mu}_0, \boldsymbol{\Sigma}_0)$, where $\boldsymbol{\mu}_0$ and $\boldsymbol{\Sigma}_0$ denote the mean vector and covariance matrix, respectively. We set the neighbourhood distribution as $\mathbb{U}(\boldsymbol{\theta}) = \mathcal{N}(\boldsymbol{\theta}|\boldsymbol{\mu}, \boldsymbol{\Sigma})$, where $\boldsymbol{\mu}$ and $\boldsymbol{\Sigma}$ denote the mean vector and covariance matrix, respectively. To further improve computational efficiency, we constrain the covariance matrix to be diagonal matrix, i.e., $\boldsymbol{\Sigma} = \text{diag}(\boldsymbol{\sigma}^2)$ and $\boldsymbol{\Sigma}_0 = \text{diag}(\boldsymbol{\rho}^2)$. We denote the density function of $\mathbb{U}(\boldsymbol{\theta})$ and $\mathbb{V}(\boldsymbol{\theta})$ as $u(\boldsymbol{\theta})$ and $v(\boldsymbol{\theta})$, respectively. We express the loss function in Eq. (8) as the following:

$$H(\mathbb{U}) = \mathbb{E}_{\boldsymbol{\theta} \sim u(\boldsymbol{\theta})}[\mathcal{L}(\boldsymbol{\mu}, \boldsymbol{\Sigma}) := -\mathcal{L}^{CL}(\boldsymbol{\theta}) + \alpha[\log u(\boldsymbol{\theta}) - \log v(\boldsymbol{\theta})]] \tag{9}$$

By parameterizing the distribution $\mathbb{U}$ as exponential family distribution in Eq. (4), our goal is to learn the parameters $\boldsymbol{\phi}$ in Eq. (5) with natural gradient descent (NGD) [42] as the following equation:

$$\boldsymbol{\phi}_{i+1} = \boldsymbol{\phi}_i - \eta \boldsymbol{F}^{-1} \nabla_{\boldsymbol{\phi}} \mathcal{L}(\boldsymbol{\phi}_i) \tag{10}$$

where $\boldsymbol{F}$ is the FIM. We opt for NGD because it adjusts the step size according to the curvature of the cost function, making convergence faster than traditional gradient descent methods. This is especially advantageous in high-dimensional spaces where the curvature and parameter-wise sensitivity vary significantly, benefiting CL models. However, computing the natural gradient is computationally intensive due to the need to calculate the FIM. To address this, we develop an efficient update method in the dual space, specifically the expectation parameter space $\boldsymbol{\lambda}$, rather than the natural parameter space $\boldsymbol{\phi}$, eliminating the need for explicit FIM calculation. In the following, we will use $\mathcal{L}(\boldsymbol{\phi})$ and $\mathcal{L}(\boldsymbol{\lambda})$ interchangeably, as they represent the same loss function only parameterized in different spaces. We leverage the relation between NGD in natural parameter space and gradient descent in expectation parameter space (in Appendix A.1), NGD can be performed without explicitly computing the FIM. This update in its dual space leads to significantly more efficient parameter updates and promising computational advantages.

**NGD for Efficiently Finding the Worst-Case Gaussian Distribution** In the following, we present specific algorithms for updating the $\boldsymbol{\mu}$ and $\boldsymbol{\Sigma}$ with NGD to find the worst-case Gaussian distribution, i.e., $\mathbb{U}^* := \arg\min_{\mathbb{U}} H(\mathbb{U})$. We can get the following updates for mean $\boldsymbol{\mu}$ and diagonal covariance $\boldsymbol{\Sigma} = \text{diag}(\boldsymbol{\sigma}^2)$ (detailed derivations can be found in Appendix B):

$$\boldsymbol{\mu}_{i+1} = \boldsymbol{\mu}_i + \eta \boldsymbol{\Sigma}_{i+1}[\nabla_{\boldsymbol{\theta}} \mathcal{L}^{CL}(\boldsymbol{\theta}) - \alpha(\boldsymbol{\mu}_i - \boldsymbol{\mu}_0)\boldsymbol{\Sigma}_0^{-1}] \tag{11}$$

$$\boldsymbol{\Sigma}_{i+1}^{-1} = (1 - \eta\alpha)\boldsymbol{\Sigma}_i^{-1} + \eta[-\nabla_{\boldsymbol{\theta}\boldsymbol{\theta}}^2 \mathcal{L}^{CL}(\boldsymbol{\theta}) + \alpha \boldsymbol{\Sigma}_0^{-1}] \tag{12}$$

By plug-in $\boldsymbol{\Sigma} = \text{diag}(\boldsymbol{\sigma}^2)$ and $\boldsymbol{\Sigma}_0 = \text{diag}(\boldsymbol{\rho}^2)$ into the above equations, we can obtain the following updates:

$$\boldsymbol{\mu}_{i+1} = \boldsymbol{\mu}_i + \eta \boldsymbol{\sigma}_{i+1}^2[\nabla_{\boldsymbol{\theta}} \mathcal{L}^{CL}(\boldsymbol{\theta}_i) - \alpha(\boldsymbol{\mu}_i - \boldsymbol{\mu}_0)\boldsymbol{\rho}^{-2}] \tag{13}$$

$$\boldsymbol{\sigma}_{i+1}^{-2} = (1 - \eta\alpha)\boldsymbol{\sigma}_i^{-2} + \eta[-\nabla_{\boldsymbol{\theta}\boldsymbol{\theta}}^2 \mathcal{L}^{CL}(\boldsymbol{\theta}_i) + \alpha\boldsymbol{\rho}^{-2}] \tag{14}$$

In practice, we set $\alpha = 1.0$ to reduce the reliance on hyperparameters. However, computing the diagonal Hessian matrix $\nabla_{\boldsymbol{\theta}\boldsymbol{\theta}}^2 \mathcal{L}^{CL}(\boldsymbol{\theta})$ in Eq. (14) is a computationally challenging task. Following [42], we efficiently approximate the Hessian as the following:

$$\nabla_{\boldsymbol{\theta}^k \boldsymbol{\theta}^k}^2 \mathcal{L}^{CL}(\boldsymbol{\theta}) = \frac{1}{N} \sum_{j=1}^{j=N} [\nabla_{\boldsymbol{\theta}^k} \mathcal{L}_j^{CL}(\boldsymbol{\theta})]^2 \tag{15}$$

where $N$ is the number of training data points for the current task, $\mathcal{L}_j^{CL}(\boldsymbol{\theta})$ denotes the loss function for the data point $j$, $\boldsymbol{\theta}^k$ denotes the $k^{th}$ element of the model parameter $\boldsymbol{\theta}$. It is crucial to note that this Hessian approximation is computed only once after learning each task and involves calculating only the diagonal elements, i.e., $\boldsymbol{\Sigma} = \text{diag}(\boldsymbol{\sigma}^2)$. As a result, the overall computational cost throughout the continual learning process remains low. Additionally, this update mechanism maintains the same number of learnable parameters as existing methods, ensuring fair comparisons. This is because, during the learning of each task, only the mean parameters of the Gaussian distribution are updated.

**Learning Algorithm** We name our method as **M**odel sensitivity **A**ware **C**ontinual **L**earning (**MACL**). The detailed algorithm is present in Algorithm 1.

**Algorithm 1** Model Sensitivity Aware Continual Learning

---

1: **REQUIRE:** model parameters $\boldsymbol{\theta}$, CL model learning rate $\beta$, worst-case Gaussian learning rate $\eta$, number of CL tasks $T$, number of CL steps $K$ for each task, distribution neighbourhood regularization strengths $\alpha = 1.0$. Randomly initialized diagonal covariance matrix, i.e., diag($\boldsymbol{\sigma}^2$).
2: **for** $n = 1$ to $T$ **do**
3:    **for** $i = 1$ to $K$ **do**
4:       calculate the CL loss function according to Eq. (1)
5:       update the worst-case Gaussian mean $\boldsymbol{\mu}$ (i.e., $\boldsymbol{\theta}$) by $\boldsymbol{\theta}'_i = \boldsymbol{\theta}_i + \eta\boldsymbol{\sigma}_n^2[\nabla_{\boldsymbol{\theta}}\mathcal{L}^{CL}(\boldsymbol{\theta}_i) - (\boldsymbol{\theta}_i - \boldsymbol{\theta}_0)\boldsymbol{\rho}^{-2}]$
6:       sample parameters from the worst-case CL model parameter distribution. $\boldsymbol{\theta}' = \boldsymbol{\theta}'_i + \boldsymbol{\sigma}_n\boldsymbol{\zeta}$, where $\boldsymbol{\zeta} \sim \mathcal{N}(\mathbf{0}, \boldsymbol{I})$
7:       update CL model parameters $\boldsymbol{\theta}_{i+1} = \boldsymbol{\theta}' - \beta\nabla_{\boldsymbol{\theta}}\mathcal{L}^{CL}(\boldsymbol{\theta}')$
8:    **end for**
9:    update the worst-case Gaussian covariance $\boldsymbol{\sigma}$ by $\boldsymbol{\sigma}_{n+1}^{-2} = (1 - \eta)\boldsymbol{\sigma}_n^{-2} + \eta[-\nabla_{\boldsymbol{\theta}\boldsymbol{\theta}}^2\mathcal{L}^{CL}(\boldsymbol{\theta}) + \boldsymbol{\rho}^{-2}]$
10:    where the Hessian is calculated by $\nabla_{\boldsymbol{\theta}^k\boldsymbol{\theta}^k}^2\mathcal{L}^{CL}(\boldsymbol{\theta}) = \frac{1}{N}\sum_{j=1}^{j=N}[\nabla_{\boldsymbol{\theta}^k}\mathcal{L}_j^{CL}(\boldsymbol{\theta})]^2$ according to Eq. (15)
11: **end for**

---

## 4 Theoretical Analysis

In this section, we build the theoretical connection between MACL and parameter sensitivity in Theorem 4.2 and the generalization analysis in Theorem 4.3. Due to the space limitations, we provide the theorem proof in Appendix A.2. Let's first look at the inner maximization problem in Eq. (7).

$$\max_{\mathbb{U}\in\mathcal{U}} \int \mathcal{L}^{CL}(\boldsymbol{\theta})d\mathbb{U}(\boldsymbol{\theta}), \quad \text{s.t. } \mathcal{U} = \{\mathbb{U} : D_{\text{KL}}(\mathbb{U}, \mathbb{V}) \leq \epsilon\} \tag{16}$$

**Lemma 4.1.** $D_{\text{KL}}(\mathbb{U}, \mathbb{V}) = \int u(\boldsymbol{\theta}) \log(\frac{u(\boldsymbol{\theta})}{v(\boldsymbol{\theta})})d\boldsymbol{\theta} \leq \int \frac{(u(\boldsymbol{\theta}) - v(\boldsymbol{\theta}))^2}{v(\boldsymbol{\theta})}d\boldsymbol{\theta}$

**Theorem 4.2.** *Assume $\int ||\frac{1}{v(\boldsymbol{\theta})}||_\infty d\boldsymbol{\theta} \leq M$, we can obtain the following conclusion for Eq. (16):*

$$\max_{\mathbb{U}\in\mathcal{U}} \int \mathcal{L}^{CL}(\boldsymbol{\theta})d\mathbb{U}(\boldsymbol{\theta}) = \overline{\mathcal{L}^{CL}(\boldsymbol{\theta})} + \sqrt{\frac{\epsilon\mathbb{E}(\mathcal{L}^{CL}(\boldsymbol{\theta}) - \overline{\mathcal{L}^{CL}(\boldsymbol{\theta})})^2}{M}} \tag{17}$$

where $\overline{\mathcal{L}^{CL}(\boldsymbol{\theta})} := \int \mathcal{L}^{CL}(\boldsymbol{\theta})d\mathbb{V}(\boldsymbol{\theta})$. $Var(\mathcal{L}^{CL}(\boldsymbol{\theta}))$ denotes the variance of $\mathcal{L}^{CL}(\boldsymbol{\theta})$ with respect to different model parameters variations, i.e., $Var(\mathcal{L}^{CL}(\boldsymbol{\theta})) = \mathbb{E}(\mathcal{L}^{CL}(\boldsymbol{\theta}) - \overline{\mathcal{L}^{CL}(\boldsymbol{\theta})})^2 = \int (\mathcal{L}^{CL}(\boldsymbol{\theta}) - \overline{\mathcal{L}^{CL}(\boldsymbol{\theta})})^2 d\boldsymbol{\theta}$. In this context, $Var(\mathcal{L}^{CL}(\boldsymbol{\theta}))$ serves as a measure of the CL model's sensitivity to parameter updates. Essentially, a smaller loss variance indicates lower parameter sensitivity in the CL model. However, directly optimizing the loss variance within the parameter distribution neighborhood is impractical, as it requires computing the loss variation across a large number of different sets of CL model parameters and training data points. In contrast, our method (MACL) offers an efficient and effective alternative. MACL implicitly minimizes the loss variance across different model parameter variations by optimizing CL performance solely on the worst-case CL model parameter distribution. In the following, inspired by UDIL [64], we further provide the following generalization bound for CL:

**Theorem 4.3** (Generalization bound of MACL). *Let $q$ be the number of CL model parameters and $n$ be the number of training data points. The CL loss $\mathcal{L}^{CL}(\boldsymbol{\theta}) \leq C$ ($C$ is a constant). With high probability of $1 - \delta$, the following bound holds:*

$$\mathbb{E}_{\boldsymbol{\theta}\sim\mathcal{N}(\boldsymbol{\mu},\boldsymbol{\Sigma})} \sum_i^{i=T} \mathcal{L}_{\mathcal{D}_i}^{CL}(\boldsymbol{\theta}) \leq \max_{\mathbb{U}\in\mathcal{U}} \mathbb{E}_{\boldsymbol{\theta}\sim\mathbb{U}}\mathcal{L}^{CL}(\boldsymbol{\theta}) + \frac{C}{N_T + \zeta\sum_{i=1}^{i=T-1} N_i} + \tag{18}$$

$$\sqrt{\frac{\tau^2(\sqrt{q} + \sqrt{2\log(N_T + \zeta\sum_{i=1}^{i=T-1} N_i)})^2 + R + 2\log(\frac{N_T + \zeta\sum_{i=1}^{i=T-1} N_i}{\delta})}{4(N_T + \zeta\sum_{i=1}^{i=T-1} N_i - 1)}}$$

*Where $\tau$ is a constant. We denote the number of data examples for task $1, \cdots, T - 1$ in the memory buffer $\mathcal{M}$ during training on task $T$ as $N_1, N_2, \cdots, N_{T-1}$ when using memory replay based approach or the number of training data points when using regularization based approach. $\mathcal{L}_{\mathcal{D}_i}^{CL}(\boldsymbol{\theta})$ denotes the CL loss on the data from data distribution $\mathcal{D}_i$ of task $i$ (generalization error), i.e., it is defined as: $\mathcal{L}_{\mathcal{D}_i}^{CL}(\boldsymbol{\theta}) := \mathbb{E}_{(\boldsymbol{x},y)\sim\mathcal{D}_i}\mathcal{L}(\boldsymbol{x}, y, \boldsymbol{\theta})$. $\mathcal{L}^{CL}(\boldsymbol{\theta})$ denotes the empirical CL loss as Eq. (1). $\mathcal{N}(\boldsymbol{\mu}, \boldsymbol{\Sigma})$ denotes the CL model parameter posterior distribution parameterized with Gaussian distribution.*

*Generalization bound implication:* (1) When using a memory-replay approach, the number of samples from new tasks often exceeds the number of samples in the memory buffer, causing data imbalance. This imbalance, where fewer samples from previous tasks are stored, affects the second and third terms in the generalization bound. The bound suggests that as the number of samples in the memory buffer increases (i.e., $\sum_{i=1}^{i=T-1} N_i \uparrow$), these terms tighten, leading to a tighter generalization upper bound. This is because $\lim_{x \to \infty}[h(x) := \frac{\log x}{x}] = 0$, meaning the generalization improves with a larger buffer, aligning with the intuition that more memory buffer data leads to better performance. (2) In the regularization-based approach, $\zeta \sum_{i=1}^{i=T-1} N_i$ is treated as the effective sample size for previous tasks since the loss is approximated in the absence of earlier data. The parameter $\zeta$ controls the trade-off between learning the new task and retaining knowledge from past tasks. A larger $\zeta$ increases regularization, preventing the model from deviating too much from the parameters learned on previous tasks. This leads to higher empirical loss on the new task (first term), but tighter bounds (second and third terms), indicating that knowledge from previous tasks is retained effectively. This prioritizes stability over learning flexibility for the new task.

## 5 Experiments

### 5.1 Setup

**Datasets** We conduct experiments on several datasets, including CIFAR10 (10 classes), CIFAR100 (100 classes) [29], and Tiny-ImageNet (200 classes) [80], to assess the effectiveness of MACL in task incremental learning (Task-IL) and class incremental learning (Class-IL). In addition, we also conduct experiments on 5-dataset [79, 5], CUB200 [69] and ImageNet-R [20] (in Appendix). Following the approach in [7], we split the CIFAR-10 dataset into five tasks, each with two distinct classes. We divided the CIFAR-100 dataset into ten tasks, each containing ten classes. We split the Tiny-ImageNet dataset into ten tasks, each comprising twenty classes. More dataset statistics can be found in Appendix E.1.

**Baselines** We compare to the following various SOTA CL methods. (1) Regularization-based methods, including oEWC [62], synaptic intelligence (SI) [84], Learning without Forgetting (LwF) [35], Classifier-Projection Regularization (CPR) [10]. (2) Bayesian-based methods, including NCL [25]. (3) Architecture-based methods, including HAT [63]. (4) Memory-based methods, including ER [15], A-GEM [14], iCaRL[55], GSS [2], HAL [12], DER++ [7], ER-ACE [8] and LODE [36]. (5) Gradient-projection-based methods: Gradient Projection Memory (GPM) [60].

**Implementation Details** Following [7], we use ResNet18 [19] as the backbone network for all the CL datasets and compared baseline methods. For the baselines that are included in the open-source code of DER++ [7], we use the same hyperparameters provided in DER++ [7] for the compared methods. For the baselines not included in the open-source code of DER++, e.g., GPM, LODE, etc, we use the open-source code from their original paper for comparisons. For the hyperparameters in our method, we set $\alpha = 1.0$ across all the datasets to minimize the model's dependence on hyperparameters. For $\eta$, we set $\eta = 1e - 5$ for CIFAR10 and CIFAR100, and $\eta = 1e - 6$ for Tiny-ImageNet. The $\eta$ is selected from the range of $[1e - 4, 1e - 5, 1e - 6, 1e - 7]$. Following [7, 14], the hyperparameter is determined through the validation sets split from the training sets from the first three tasks. Similar to [7], we train all the CL models using the standard SGD optimizer to update the CL model. The batch size and replay buffer batch size are set to 32. We use a single NVIDIA A5000 GPU with 24GB memory to run the experiments. Each experiment result is averaged for 10 runs with mean and standard deviation.

### 5.2 Results

We evaluate the performance of different CL methods with (1) overall accuracy; (2) new task accuracy; and (3) backward transfer in the following.

**Overall Accuracy (ACC)** ACC is the average accuracy across the entire task sequence. We present the results on CIFAR10, CIFAR-100 and Tiny-ImageNet in Table 1. We can observe that our method substantially improve over various SOTA baseline methods up to 3% to 4% on CIFAR100, TinyImageNet by integrating MACL with existing CL methods. This overall performance improvement is attributed to the reduced parameter sensitivity.

Table 1: **Task-IL and class-IL** overall accuracy on CIFAR10, CIFAR-100 and Tiny-ImageNet, respectively with memory size 500. '—' indicates not applicable/available.

| Method | CIFAR-10 | | CIFAR-100 | | Tiny-ImageNet | |
|---|---|---|---|---|---|---|
| | Class-IL | Task-IL | Class-IL | Task-IL | Class-IL | Task-IL |
| fine-tuning | $19.62 \pm 0.05$ | $61.02 \pm 3.33$ | $9.29 \pm 0.33$ | $33.78 \pm 0.42$ | $7.92 \pm 0.26$ | $18.31 \pm 0.68$ |
| Joint train | $92.20 \pm 0.15$ | $98.31 \pm 0.12$ | $71.32 \pm 0.21$ | $91.31 \pm 0.17$ | $59.99 \pm 0.19$ | $82.04 \pm 0.10$ |
| SI | $19.48 \pm 0.17$ | $68.05 \pm 5.91$ | $9.41 \pm 0.24$ | $31.08 \pm 1.65$ | $6.58 \pm 0.31$ | $36.32 \pm 0.13$ |
| LwF | $19.61 \pm 0.05$ | $63.29 \pm 2.35$ | $9.70 \pm 0.23$ | $28.07 \pm 1.96$ | $8.46 \pm 0.22$ | $15.85 \pm 0.58$ |
| NCL | $19.53 \pm 0.32$ | $64.49 \pm 4.06$ | $8.12 \pm 0.28$ | $20.92 \pm 2.32$ | $7.56 \pm 0.36$ | $16.29 \pm 0.87$ |
| GPM | —— | $90.68 \pm 3.29$ | —— | $72.48 \pm 0.40$ | —— | —— |
| UCB | —— | $79.28 \pm 1.87$ | —— | $57.15 \pm 1.67$ | —— | —— |
| HAT | —— | $92.56 \pm 0.78$ | —— | $72.06 \pm 0.50$ | —— | —— |
| A-GEM | $22.67 \pm 0.57$ | $89.48 \pm 1.45$ | $9.30 \pm 0.32$ | $48.06 \pm 0.57$ | $8.06 \pm 0.04$ | $25.33 \pm 0.49$ |
| GSS | $49.73 \pm 4.78$ | $91.02 \pm 1.57$ | $13.60 \pm 2.98$ | $57.50 \pm 1.93$ | —— | —— |
| HAL | $41.79 \pm 4.46$ | $84.54 \pm 2.36$ | $9.05 \pm 2.76$ | $42.94 \pm 1.80$ | —— | —— |
| oEWC | $19.49 \pm 0.12$ | $64.31 \pm 4.31$ | $8.24 \pm 0.21$ | $21.2 \pm 2.08$ | $7.42 \pm 0.31$ | $15.19 \pm 0.82$ |
| oEWC+MACL | $\mathbf{20.55 \pm 0.71}$ | $\mathbf{66.95 \pm 2.46}$ | $\mathbf{8.82 \pm 0.50}$ | $\mathbf{23.42 \pm 1.93}$ | $\mathbf{7.86 \pm 0.23}$ | $\mathbf{17.43 \pm 0.93}$ |
| CPR(EWC) | $19.61 \pm 3.67$ | $65.23 \pm 3.87$ | $8.42 \pm 0.37$ | $21.43 \pm 2.57$ | $7.67 \pm 0.23$ | $15.58 \pm 0.91$ |
| CPR(EWC)+MACL | $\mathbf{20.58 \pm 2.56}$ | $\mathbf{67.28 \pm 3.75}$ | $\mathbf{9.15 \pm 0.63}$ | $\mathbf{22.87 \pm 1.78}$ | $\mathbf{8.10 \pm 0.49}$ | $\mathbf{17.96 \pm 0.82}$ |
| GPM | —— | —— | —— | $72.48 \pm 0.40$ | —— | $30.72 \pm 0.27$ |
| GPM+MACL | —— | —— | —— | $\mathbf{74.51 \pm 0.36}$ | —— | $\mathbf{35.06 \pm 0.38}$ |
| iCaRL | —— | —— | $44.16 \pm 1.53$ | $84.06 \pm 0.42$ | $23.71 \pm 0.23$ | $59.24 \pm 0.16$ |
| iCaRL+MACL | —— | —— | $\mathbf{48.27 \pm 0.95}$ | $\mathbf{84.55 \pm 0.51}$ | $\mathbf{24.18 \pm 0.58}$ | $\mathbf{59.45 \pm 0.32}$ |
| ER | $57.74 \pm 0.27$ | $93.61 \pm 0.27$ | $20.98 \pm 0.35$ | $73.37 \pm 0.43$ | $\mathbf{9.99 \pm 0.29}$ | $48.64 \pm 0.46$ |
| ER+MACL | $\mathbf{63.74 \pm 1.24}$ | $\mathbf{93.78 \pm 0.36}$ | $\mathbf{22.18 \pm 0.27}$ | $\mathbf{74.87 \pm 0.51}$ | $9.87 \pm 0.15$ | $\mathbf{51.25 \pm 0.37}$ |
| DER++ | $72.70 \pm 1.36$ | $93.88 \pm 0.50$ | $36.37 \pm 0.85$ | $75.64 \pm 0.60$ | $19.38 \pm 1.41$ | $51.91 \pm 0.68$ |
| DER+++MACL | $\mathbf{74.53 \pm 0.79}$ | $\mathbf{94.72 \pm 0.65}$ | $\mathbf{39.42 \pm 0.82}$ | $\mathbf{77.53 \pm 0.89}$ | $\mathbf{20.17 \pm 1.56}$ | $\mathbf{54.03 \pm 0.79}$ |
| ER-ACE | $71.83 \pm 1.42$ | $94.12 \pm 0.61$ | $37.05 \pm 0.36$ | $75.97 \pm 0.69$ | $20.43 \pm 0.97$ | $52.59 \pm 0.75$ |
| ER-ACE+MACL | $\mathbf{73.21 \pm 0.96}$ | $\mathbf{94.98 \pm 0.72}$ | $\mathbf{40.28 \pm 0.39}$ | $\mathbf{77.65 \pm 0.76}$ | $\mathbf{21.89 \pm 0.83}$ | $\mathbf{53.95 \pm 0.78}$ |
| LODE | $75.45 \pm 0.90$ | $\mathbf{94.41 \pm 0.22}$ | $38.95 \pm 0.93$ | $78.92 \pm 0.67$ | $19.87 \pm 0.72$ | $60.18 \pm 0.65$ |
| LODE+MACL | $\mathbf{76.41 \pm 0.67}$ | $94.32 \pm 0.24$ | $\mathbf{40.67 \pm 0.89}$ | $40.03 \pm 0.51$ | $\mathbf{21.09 \pm 0.97}$ | $\mathbf{61.79 \pm 0.86}$ |

**New Task Accuracy** To evaluate the new task performance of the proposed CL method, we evaluate the new task performance during CL by integrating MACL with DER++ and GPM in Figure 1. The results show that MACL can significantly improves the new task performance for different CL methods, indicating that reducing the model parameter sensitivity is beneficial to improve new task performance during CL.

**Backward Transfer** Backward transfer (BWT) quantifies the degree of forgetting observed on previously learned tasks. When BWT $> 0$, it indicates that learning the current new task positively influences the performance on previously learned tasks. Conversely, when BWT $\leq 0$, it signals that learning the current new task may result in forgetting previously acquired knowledge. We evaluate BWT in Table 2. We can observe that our method significantly improves BWT by up to 5% through integrating MACL with existing CL methods. This indicates that reducing parameter sensitivity can substantially reduce forgetting on previously learned knowledge. These empirical analysis also verify our theoretical analysis that our method implicitly improves the stability by reducing loss variance.

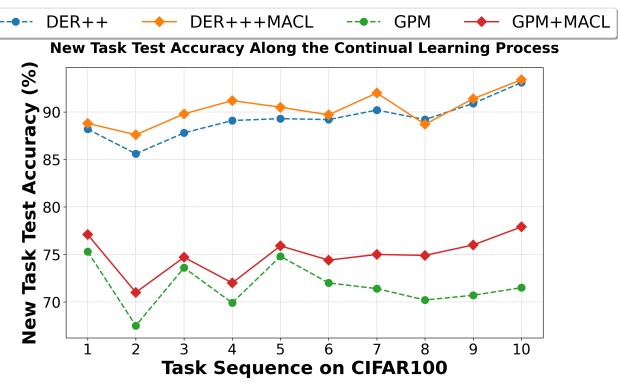

Figure 1: new task performance during CL.

Table 2: **Backward Transfer** of different CL methods with memory size 500.

| Method | CIFAR10 | | CIFAR100 | | Tiny-ImageNet | |
|---|---|---|---|---|---|---|
| | Class-IL | Task-IL | Class-IL | Task-IL | Class-IL | Task-IL |
| finetuning | $-96.39 \pm 0.12$ | $-46.24 \pm 2.12$ | $-89.68 \pm 0.96$ | $-62.46 \pm 0.78$ | $-78.94 \pm 0.81$ | $-67.34 \pm 0.79$ |
| AGEM | $-94.01 \pm 1.16$ | $-14.26 \pm 1.18$ | $-88.5 \pm 1.56$ | $-45.43 \pm 2.32$ | $-78.03 \pm 0.78$ | $-59.28 \pm 1.08$ |
| GSS | $-62.88 \pm 2.67$ | $-7.73 \pm 3.99$ | $-82.17 \pm 4.16$ | $-33.98 \pm 1.54$ | —— | —— |
| HAL | $-62.21 \pm 4.34$ | $-5.41 \pm 1.10$ | $-49.29 \pm 2.82$ | $-13.60 \pm 1.04$ | —— | —— |
| ER | $-45.35 \pm 0.07$ | -3.54 $\pm$ 0.35 | $-74.84 \pm 1.38$ | $-16.81 \pm 0.97$ | **-75.24 $\pm$ 0.76** | $-31.98 \pm 1.35$ |
| ER+MACL | **-34.43 $\pm$ 0.82** | **-3.31 $\pm$ 0.32** | **-73.17 $\pm$ 0.69** | **-15.73 $\pm$ 0.78** | $-75.29 \pm 0.37$ | **-29.32 $\pm$ 0.42** |
| DER++ | $-22.38 \pm 4.41$ | $-4.66 \pm 1.15$ | $-53.89 \pm 1.85$ | $-14.72 \pm 0.96$ | $-64.6 \pm 0.56$ | $-27.21 \pm 1.23$ |
| DER++ MACL | **-21.87 $\pm$ 1.67** | **-3.09 $\pm$ 1.31** | **-48.62 $\pm$ 1.56** | **-13.62 $\pm$ 0.35** | **-62.23 $\pm$ 0.78** | **-27.10 $\pm$ 0.43** |
| ER-ACE | -13.64 $\pm$ 0.95 | -3.28 $\pm$ 0.83 | -39.51 $\pm$ 1.23 | -14.57 $\pm$ 0.39 | -46.07 $\pm$ 0.83 | **-28.35 $\pm$ 0.16** |
| ER-ACE+MACL | **-12.76 $\pm$ 1.23** | **-3.15 $\pm$ 0.57** | **-33.86 $\pm$ 1.37** | **-13.89 $\pm$ 0.57** | **-42.29 $\pm$ 0.50** | -28.41 $\pm$ 0.23 |
| LODE | -16.37 $\pm$ 0.67 | **-2.93 $\pm$ 0.19** | -53.23 $\pm$ 1.72 | -15.24 $\pm$ 0.76 | -55.89 $\pm$ 0.98 | -19.13 $\pm$ 0.56 |
| LODE+MACL | **-16.25 $\pm$ 0.73** | -3.16 $\pm$ 0.45 | **-52.67 $\pm$ 1.35** | **-15.11 $\pm$ 0.53** | **-55.61 $\pm$ 1.15** | **-18.17 $\pm$ 0.83** |

## 5.3 Ablation Study

**Hyperparameter Analysis** We evaluate the sensitivity of the hyperparameters $\eta$ in Table 5 in Appendix D.1. Our observations indicate that when parameter sensitivity is not reduced, i.e., $\eta = 0$, the CL model performs poorly. As we gradually increase the reduction of parameter sensitivity, the CL model's performance improves. This improvement is because appropriately reducing parameter sensitivity helps mitigate forgetting and enhances learning for new tasks, thus boosting overall CL performance. However, if the reduction in parameter sensitivity is increased excessively, the model's performance deteriorates. This is because an overly constrained model, while minimizing forgetting, struggles to learn new tasks effectively, resulting in worse performance.

**Effect of Memory Size** To assess the impact of varying memory buffer sizes, we present the results in Table 3. The results demonstrate that compared to different baseline methods, our MACL plug-in also enhances the performance of baseline methods with a memory size of 2000.

Table 3: **Task-IL and class-IL** overall accuracy on CIFAR-100 and Tiny-ImageNet, respectively with memory size 2000.

| Algorithm Method | CIFAR-100 | | Tiny-ImageNet | |
|---|---|---|---|---|
| | Class-IL | Task-IL | Class-IL | Task-IL |
| ER | $36.06 \pm 0.72$ | $81.09 \pm 0.45$ | $15.16 \pm 0.78$ | $58.19 \pm 0.69$ |
| ER+MACL | **37.83 $\pm$ 0.94** | **83.37 $\pm$ 1.35** | **17.08 $\pm$ 0.73** | **59.51 $\pm$ 0.53** |
| DER++ | $50.72 \pm 0.71$ | $82.43 \pm 0.38$ | $24.21 \pm 1.09$ | $62.22 \pm 0.87$ |
| DER+++MACL | **52.79 $\pm$ 0.85** | **84.07 $\pm$ 0.79** | **27.55 $\pm$ 1.43** | **64.28 $\pm$ 0.95** |
| LODE | $54.32 \pm 0.56$ | $85.79 \pm 0.67$ | $31.03 \pm 1.27$ | **70.05 $\pm$ 0.59** |
| LODE+MACL | **54.76 $\pm$ 0.68** | **86.53 $\pm$ 0.58** | **32.16 $\pm$ 1.12** | $69.79 \pm 0.53$ |

**Benefit of NGD** To evaluate the benefits of using NGD over gradient descent (GD) for calculating the worst-case Gaussian distribution, we present comparison results in Table 6 in Appendix D.2. The results show that NGD outperforms GD because NGD better captures parameter importance, which helps preserve old knowledge while effectively adapting to new tasks.

**Efficiency Evaluation** To assess the efficiency of our proposed method, we compare the running time of integration of different CL methods with MACL and corresponding CL methods alone on CIFAR100, as shown in Table 15 in Appendix D.8. The results indicate that incorporating MACL increases the computational cost by only 55% to 61% compared to the corresponding CL methods alone. This demonstrates the high efficiency of our method, as it introduces only small additional training cost.

**Effect of Different Architectures** To evaluate the impact of different architectures, we compared various approaches using both ViT and ResNet32. For the ResNet32 experiments, we followed the setup in [85], integrating MACL with MEMO [86] and comparing it to MEMO alone, using a memory buffer size of 2000 on CIFAR100. Additionally, we conducted experiments with a pre-trained Vision Transformer (ViT) [16], specifically the vit-base-patch16-224 model pre-trained on ImageNet1K. On CIFAR100, we integrated MACL with DER++, using a memory size of 500, and demonstrated that using a pre-trained ViT significantly improves CL performance. Moreover, combining MACL with DER++ further enhances CL performance with the pre-trained ViT. The results are presented in the Appendix.

**Long Task Sequence** To assess the effectiveness of the proposed approach across varying task lengths, we conducted experiments by splitting Tiny-ImageNet into sequences of 10 and 20 tasks. The Task-IL and Class-IL results for integrating DER++ with MACL, using a memory buffer size of 500, are presented in Table 4. These results demonstrate that even with longer task sequences, our method still significantly outperforms DER++.

Table 4: Overall accuracy of integrating DER++ with MACL using a memory buffer of 500 and longer task sequence on Tiny-ImageNet.

| number of tasks | 10 | 20 |
|---|---|---|
| Class-IL | $19.38 \pm 1.41$ | $15.02 \pm 0.53$ |
| Class-IL+ MACL | $\mathbf{20.17 \pm 1.56}$ | $\mathbf{16.08 \pm 0.81}$ |
| Task-IL | $51.91 \pm 0.68$ | $51.65 \pm 1.36$ |
| Task-IL + MACL | $\mathbf{54.03 \pm 0.79}$ | $\mathbf{54.96 \pm 0.72}$ |

**Online CL** Under the online CL setting, we evaluate the effectiveness of the proposed approach on CIFAR100 and Tiny-ImageNet by comparing with MKD [46] and PCR [37]. The results are put in the Appendix.

**5-datasets results** To assess the effectiveness of MACL on the 5-Datasets benchmark [79, 5], which includes CIFAR-10, MNIST [33], Fashion-MNIST [81], SVHN [47], and notMNIST [6], we conducted experiments. This dataset provides a diverse range of CL tasks. We performed experiments on 5-Datasets, using a memory buffer size of 500, with MACL. The detailed results are provided in the Appendix.

**ImageNet-R and CUB200 results** We further evaluate the effectiveness of MACL on CUB200 [69] and ImageNet-R [20], the results are shown in the Appendix.

## 6 Conclusion

In this paper, we address the challenge of balancing learning new tasks while preserving knowledge from previous ones in continual learning. We propose a model sensitivity-aware continual learning method that enhances both the model's ability to retain old knowledge and improve performance on new tasks. Specifically, our goal is to reduce model parameter sensitivity by optimizing CL performance for the worst-case parameter distribution within the neighborhood of the current model's parameter distribution. This approach improves stability in preserving old knowledge and mitigates overfitting on new tasks. We provide a comprehensive theoretical analysis of the proposed method, and extensive experiments on multiple datasets demonstrate its effectiveness, efficiency, and versatility.

**Limitation Discussion** Our method introduces additional training cost compared to existing continual learning approaches.

## Broader Impacts

Our work advances continual learning, which is beneficial to develop more adaptable and efficient AI. Our work has no negative societal impacts.

## Acknowledgments

This work was partially supported by NSF IIS 2347592, 2347604, 2348159, 2348169, DBI 2405416, CCF 2348306, CNS 2347617.

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

# A  Theorem Proof

## A.1  Duality in Natural Gradient Descent for Exponential Family Distribution

**Theorem A.1.** *Gradient of the loss $\mathcal{L}(\boldsymbol{\lambda})$ with respect to the expectation parameter $\boldsymbol{\lambda}$, i.e., $\nabla_{\boldsymbol{\lambda}}\mathcal{L}(\boldsymbol{\lambda})$, is equal to the natural gradient with respect to natural parameter $\boldsymbol{\phi}$, i.e., $\boldsymbol{F}^{-1}\nabla_{\boldsymbol{\phi}}\mathcal{L}(\boldsymbol{\phi})$. This can be expressed as the following:*

$$\nabla_{\boldsymbol{\lambda}}\mathcal{L}(\boldsymbol{\lambda}) = \boldsymbol{F}^{-1}\nabla_{\boldsymbol{\phi}}\mathcal{L}(\boldsymbol{\phi}) \tag{19}$$

*In particular, NGD in natural parameter space can be equivalently performed through gradient descent with respect to the expectation parameters as the following:*

$$\boldsymbol{\phi}_{i+1} = \boldsymbol{\phi}_i - \eta\boldsymbol{F}^{-1}\nabla_{\boldsymbol{\phi}}\mathcal{L}(\boldsymbol{\phi}_i) = \boldsymbol{\phi}_i - \eta\nabla_{\boldsymbol{\lambda}}\mathcal{L}(\boldsymbol{\lambda}_i) \tag{20}$$

*where $\eta$ is the learning rate and $\boldsymbol{F}$ is the Fisher information matrix (FIM).*

*Proof.* The exponential family distribution is defined as the following:

$$P_{\boldsymbol{\phi}}(\boldsymbol{\theta}) = exp(\langle\boldsymbol{\phi}, \boldsymbol{\Omega}(\boldsymbol{\theta})\rangle - Z(\boldsymbol{\phi})) \tag{21}$$

According to the expectation of the score function is 0, we can obtain the following

$$\boldsymbol{0} = \mathbb{E}_{P_{\boldsymbol{\phi}}(\boldsymbol{\theta})}\nabla_{\boldsymbol{\phi}}\log P_{\boldsymbol{\phi}}(\boldsymbol{\theta}) = \mathbb{E}_{P_{\boldsymbol{\phi}}(\boldsymbol{\theta})}[\boldsymbol{\Omega}(\boldsymbol{\theta}) - \nabla_{\boldsymbol{\phi}}Z(\boldsymbol{\phi})] = \boldsymbol{\lambda} - \nabla_{\boldsymbol{\phi}}Z(\boldsymbol{\phi}) \tag{22}$$

Therefore,

$$\boldsymbol{\lambda} = \nabla_{\boldsymbol{\phi}}Z(\boldsymbol{\phi}) \tag{23}$$

where the first equality is due to the fact that the expectation of the score function is zero.

We then derive the Fisher information matrix (FIM) as the following:

$$\boldsymbol{F}(\boldsymbol{\phi}) := \mathbb{E}_{P_{\boldsymbol{\phi}}(\boldsymbol{\theta})}[-\nabla_{\boldsymbol{\phi}}^2\log P_{\boldsymbol{\phi}}(\boldsymbol{\theta})] \tag{24}$$

$$= \mathbb{E}_{P_{\boldsymbol{\phi}}(\boldsymbol{\theta})}[-\nabla_{\boldsymbol{\phi}}(\nabla_{\boldsymbol{\phi}}\log P_{\boldsymbol{\phi}}(\boldsymbol{\theta}))] \tag{25}$$

$$= \mathbb{E}_{P_{\boldsymbol{\phi}}(\boldsymbol{\theta})}[-\nabla_{\boldsymbol{\phi}}(\nabla_{\boldsymbol{\phi}}(\langle\boldsymbol{\phi}, \boldsymbol{\Omega}(\boldsymbol{\theta})\rangle - Z(\boldsymbol{\phi})))] \tag{26}$$

$$= \mathbb{E}_{P_{\boldsymbol{\phi}}(\boldsymbol{\theta})}[-\nabla_{\boldsymbol{\phi}}(\boldsymbol{\Omega}(\boldsymbol{\theta}) - \nabla_{\boldsymbol{\phi}}Z(\boldsymbol{\phi}))] \tag{27}$$

$$= \nabla_{\boldsymbol{\phi}}\boldsymbol{\lambda} \tag{28}$$

$$= \nabla_{\boldsymbol{\phi}}\nabla_{\boldsymbol{\phi}}Z(\boldsymbol{\phi}) \tag{29}$$

$$= \nabla_{\boldsymbol{\phi}}^2 Z(\boldsymbol{\phi}) \tag{30}$$

where := denotes defined as. Then,

$$\nabla_{\boldsymbol{\phi}}\boldsymbol{\lambda} = \nabla_{\boldsymbol{\phi}}^2 Z(\boldsymbol{\phi}) = \boldsymbol{F} \tag{31}$$

Next,

$$\nabla_{\boldsymbol{\lambda}}\mathcal{L}(\boldsymbol{\phi}) = \nabla_{\boldsymbol{\lambda}}\boldsymbol{\phi}\nabla_{\boldsymbol{\phi}}\mathcal{L}(\boldsymbol{\phi}) = [\nabla_{\boldsymbol{\phi}}\boldsymbol{\lambda}]^{-1}\nabla_{\boldsymbol{\phi}}\mathcal{L}(\boldsymbol{\phi}) = \boldsymbol{F}^{-1}\nabla_{\boldsymbol{\phi}}\mathcal{L}(\boldsymbol{\phi}) \tag{32}$$

$\square$

More general results on manifold can be found in [53].

## A.2  Theoretical and Generalization Analysis of MACL

**Lemma A.2.** $D_{\mathrm{KL}}(\mathbb{U}, \mathbb{V}) = \int u(\boldsymbol{\theta})\log(\frac{u(\boldsymbol{\theta})}{v(\boldsymbol{\theta})})d\boldsymbol{\theta} \le \int \frac{(u(\boldsymbol{\theta})-v(\boldsymbol{\theta}))^2}{v(\boldsymbol{\theta})}d\boldsymbol{\theta}$

*Proof.*

$$D_{\mathrm{KL}}(\mathbb{U}, \mathbb{V}) = \int u(\boldsymbol{\theta}) \log(\frac{u(\boldsymbol{\theta})}{v(\boldsymbol{\theta})}) d\boldsymbol{\theta} \tag{33}$$

$$\leq \log \int \frac{u(\boldsymbol{\theta})^2}{v(\boldsymbol{\theta})} d\boldsymbol{\theta} \quad \text{(by Jensen's inequality)} \tag{34}$$

$$\leq \int \frac{u(\boldsymbol{\theta})^2}{v(\boldsymbol{\theta})} - 1 d\boldsymbol{\theta} \quad (\log(1+\boldsymbol{x}) \leq \boldsymbol{x}) \tag{35}$$

$$= \int \frac{(u(\boldsymbol{\theta}) - v(\boldsymbol{\theta}))^2}{v(\boldsymbol{\theta})} d\boldsymbol{\theta} \tag{36}$$

where the last equality is because

$$\int \frac{(u(\boldsymbol{\theta}) - v(\boldsymbol{\theta}))^2}{v(\boldsymbol{\theta})} d\boldsymbol{\theta} = \int \frac{u(\boldsymbol{\theta})^2}{v(\boldsymbol{\theta})} - 2\int u(\boldsymbol{\theta}) d\boldsymbol{\theta} + \int v(\boldsymbol{\theta}) d\boldsymbol{\theta} = \int \frac{u(\boldsymbol{\theta})^2}{v(\boldsymbol{\theta})} - 1 \tag{37}$$

Since $\int u(\boldsymbol{\theta}) d\boldsymbol{\theta} = \int v(\boldsymbol{\theta}) d\boldsymbol{\theta} = 1$

$\square$

**Theorem A.3.** *Assume $\int ||\frac{1}{v(\boldsymbol{\theta})}||_{\infty} d\boldsymbol{\theta} \leq M$, we can obtain the following conclusion for Eq. (16):*

$$\max_{\mathbb{U} \in \mathcal{U}} \int \mathcal{L}^{CL}(\boldsymbol{\theta}) d\mathbb{U}(\boldsymbol{\theta}) = \overline{\mathcal{L}^{CL}(\boldsymbol{\theta})} + \sqrt{\frac{\epsilon \mathbb{E}(\mathcal{L}^{CL}(\boldsymbol{\theta}) - \overline{\mathcal{L}^{CL}(\boldsymbol{\theta})})^2}{M}} \tag{38}$$

*where $\overline{\mathcal{L}^{CL}(\boldsymbol{\theta})} := \int \mathcal{L}^{CL}(\boldsymbol{\theta}) d\mathbb{V}(\boldsymbol{\theta})$. We denote the variance of the random variable $\mathcal{L}^{CL}(\boldsymbol{\theta})$ as $Var(\mathcal{L}^{CL}(\boldsymbol{\theta})) = \mathbb{E}(\mathcal{L}^{CL}(\boldsymbol{\theta}) - \overline{\mathcal{L}^{CL}(\boldsymbol{\theta})})^2 = \int (\mathcal{L}^{CL}(\boldsymbol{\theta}) - \overline{\mathcal{L}^{CL}(\boldsymbol{\theta})})^2 d\boldsymbol{\theta}$.*

*Proof.* We define a new distribution $\mathbb{Z} := \mathbb{U} - \mathbb{V}$.

$$\int \mathcal{L}^{CL}(\boldsymbol{\theta}) d\mathbb{U}(\boldsymbol{\theta}) = \int \mathcal{L}^{CL}(\boldsymbol{\theta}) d(\mathbb{Z}(\boldsymbol{\theta}) + \mathbb{V}(\boldsymbol{\theta})) = \overline{\mathcal{L}^{CL}(\boldsymbol{\theta})} + \int \mathcal{L}^{CL}(\boldsymbol{\theta}) d\mathbb{Z}(\boldsymbol{\theta}) \tag{39}$$

$$= \overline{\mathcal{L}^{CL}(\boldsymbol{\theta})} + \int (\mathcal{L}^{CL}(\boldsymbol{\theta}) - \overline{\mathcal{L}^{CL}(\boldsymbol{\theta})}) d\mathbb{Z}(\boldsymbol{\theta}) + \int \overline{\mathcal{L}^{CL}(\boldsymbol{\theta})} d\mathbb{Z}(\boldsymbol{\theta}) \tag{40}$$

By Lemma 4.1 and Hölder's inequality, we can obtain the following:

$$D_{\mathrm{KL}}(\mathbb{U}, \mathbb{V}) = \int u(\boldsymbol{\theta}) \log(\frac{u(\boldsymbol{\theta})}{v(\boldsymbol{\theta})}) d\boldsymbol{\theta} \leq \int \frac{(u(\boldsymbol{\theta}) - v(\boldsymbol{\theta}))^2}{v(\boldsymbol{\theta})} d\boldsymbol{\theta} \tag{41}$$

$$\leq \int (u(\boldsymbol{\theta}) - v(\boldsymbol{\theta}))^2 d\boldsymbol{\theta} \int ||\frac{1}{v(\boldsymbol{\theta})}||_{\infty} d\boldsymbol{\theta} \tag{42}$$

$$\leq \int (u(\boldsymbol{\theta}) - v(\boldsymbol{\theta}))^2 d\boldsymbol{\theta} M \leq \epsilon \tag{43}$$

Therefore,

$$\int (u(\boldsymbol{\theta}) - v(\boldsymbol{\theta}))^2 d\boldsymbol{\theta} \leq \frac{\epsilon}{M} \tag{44}$$

$$\int (\mathcal{L}^{CL}(\boldsymbol{\theta}) - \overline{\mathcal{L}^{CL}(\boldsymbol{\theta})}) d\mathbb{Z}(\boldsymbol{\theta}) = \int (\mathcal{L}^{CL}(\boldsymbol{\theta}) - \overline{\mathcal{L}^{CL}(\boldsymbol{\theta})})(u(\boldsymbol{\theta}) - v(\boldsymbol{\theta})) d\boldsymbol{\theta} \tag{45}$$

$$\leq \sqrt{\int (\mathcal{L}^{CL}(\boldsymbol{\theta}) - \overline{\mathcal{L}^{CL}(\boldsymbol{\theta})})^2 d\boldsymbol{\theta} \int (u(\boldsymbol{\theta}) - v(\boldsymbol{\theta}))^2 d\boldsymbol{\theta}} \quad \text{(Cauchy-Schwarz inequality)} \tag{46}$$

$$\leq \sqrt{\frac{\epsilon \mathbb{E}(\mathcal{L}^{CL}(\boldsymbol{\theta}) - \overline{\mathcal{L}^{CL}(\boldsymbol{\theta})})^2}{M}} \tag{47}$$

The equality holds when the following condition holds:

$$u(\boldsymbol{\theta}) - v(\boldsymbol{\theta}) = a(\mathcal{L}^{CL}(\boldsymbol{\theta}) - \overline{\mathcal{L}^{CL}(\boldsymbol{\theta})}) \tag{48}$$

where $a$ is a constant.

$$\int \overline{\mathcal{L}^{CL}(\boldsymbol{\theta})} d\mathbb{Z}(\boldsymbol{\theta}) = \int \overline{\mathcal{L}^{CL}(\boldsymbol{\theta})}(u(\boldsymbol{\theta}) - v(\boldsymbol{\theta})) d\boldsymbol{\theta} \tag{49}$$

$$= \overline{\mathcal{L}^{CL}(\boldsymbol{\theta})} \int (u(\boldsymbol{\theta}) - v(\boldsymbol{\theta})) d\boldsymbol{\theta} \tag{50}$$

$$= 0 \tag{51}$$

The last equality is because $\int (u(\boldsymbol{\theta}) - v(\boldsymbol{\theta})) d\boldsymbol{\theta} = \int u(\boldsymbol{\theta}) d\boldsymbol{\theta} - \int v(\boldsymbol{\theta}) d\boldsymbol{\theta} = 1 - 1 = 0$

Therefore, we can obtain the following conclusion:

$$\max_{\mathbb{U} \in \mathcal{U}} \int \mathcal{L}^{CL}(\boldsymbol{\theta}) d\mathbb{U}(\boldsymbol{\theta}) = \overline{\mathcal{L}^{CL}(\boldsymbol{\theta})} + \sqrt{\frac{\epsilon \mathbb{E}(\mathcal{L}^{CL}(\boldsymbol{\theta}) - \overline{\mathcal{L}^{CL}(\boldsymbol{\theta})})^2}{M}} \tag{52}$$

$\square$

In this context, the CL loss variance across various sets of model parameters $Var(\mathcal{L}^{CL}(\boldsymbol{\theta}))$ serves as a measure of the CL model's sensitivity to parameter updates. Essentially, a smaller loss variance indicates lower parameter sensitivity in the CL model. However, directly optimizing the loss variance within the parameter distribution neighborhood is impractical, as it requires computing the loss variance across a large number of different sets of CL model parameters and training data points. In contrast, our method (MACL) offers an efficient and effective alternative. MACL implicitly minimizes the loss variance across different model parameter variations by optimizing CL performance solely on the worst-case CL model parameter distribution.

We denote the prior distribution as $\mathbb{V}(\boldsymbol{\theta}) = \mathcal{N}(\boldsymbol{\mu}_p, \boldsymbol{\Sigma}_p)$ and posterior distribution as $\mathbb{U}(\boldsymbol{\theta}) = \mathcal{N}(\boldsymbol{\mu}_s, \boldsymbol{\Sigma}_s)$

**Theorem A.4** (Generalization bound of MACL). *Let $q$ be the number of CL model parameters and $n$ be the number of training data points. The CL loss $\mathcal{L}^{CL}(\boldsymbol{\theta}) \leq C$ ($C$ is a constant). With high probability of $1 - \delta$, the following bound holds:*

$$\mathbb{E}_{\boldsymbol{\theta} \sim \mathcal{N}(\boldsymbol{\mu}, \boldsymbol{\Sigma})} \sum_i^{i=T} \mathcal{L}_{\mathcal{D}_i}^{CL}(\boldsymbol{\theta}) \leq \max_{\mathbb{U} \in \mathcal{U}} \mathbb{E}_{\boldsymbol{\theta} \sim \mathbb{U}} \mathcal{L}^{CL}(\boldsymbol{\theta}) + \frac{C}{N_T + \zeta \sum_{i=1}^{i=T-1} N_i} + \tag{53}$$

$$\sqrt{\frac{\tau^2 (\sqrt{q} + \sqrt{2 \log(N_T + \zeta \sum_{i=1}^{i=T-1} N_i)})^2 + R + 2 \log(\frac{N_T + \zeta \sum_{i=1}^{i=T-1} N_i}{\delta})}{4(N_T + \zeta \sum_{i=1}^{i=T-1} N_i - 1)}}$$

*Where $\tau$ is a constant. We denote the number of data examples for task $1, \cdots, T - 1$ in the memory buffer $\mathcal{M}$ during training on task $T$ as $N_1, N_2, \cdots, N_{T-1}$ when using memory replay based approach or the number of training data points when using regularization based approach. $\mathcal{L}_{\mathcal{D}_i}^{CL}(\boldsymbol{\theta})$ denotes the CL loss on the data from data distribution $\mathcal{D}_i$ (generalization error), i.e., it is defined as: $\mathcal{L}_{\mathcal{D}_i}^{CL}(\boldsymbol{\theta}) := \mathbb{E}_{(\boldsymbol{x},y) \sim \mathcal{D}_i} \mathcal{L}(\boldsymbol{x}, y, \boldsymbol{\theta})$. $\mathcal{L}^{CL}(\boldsymbol{\theta})$ denotes the empirical CL loss as Eq. (1). $\mathcal{N}(\boldsymbol{\mu}, \boldsymbol{\Sigma})$ denotes the CL model parameter posterior distribution parameterized with Gaussian distribution.*

*Proof.* We apply the PAC-Bayes theorem [43] that for any prior distribution, with probability $1 - \delta$ over the CL training dataset $\mathcal{T}$, the following bound holds:

$$\mathbb{E}_{\boldsymbol{\theta} \sim \mathbb{U}(\boldsymbol{\theta})} [\mathcal{L}_{\mathcal{D}}^{CL}(\boldsymbol{\theta})] \leq \mathbb{E}_{\boldsymbol{\theta} \sim \mathbb{U}(\boldsymbol{\theta})} [\mathcal{L}_{\mathcal{T}}^{CL}(\boldsymbol{\theta})] + \sqrt{\frac{D_{\mathrm{KL}}(\mathbb{U}(\boldsymbol{\theta}) || \mathbb{V}(\boldsymbol{\theta})) + \log(\frac{n}{\delta})}{2(n-1)}} \tag{54}$$

The KL divergence between posterior and prior distribution can be calculated as the following:

$$D_{\mathrm{KL}}(\mathbb{U}(\boldsymbol{\theta})||\mathbb{V}(\boldsymbol{\theta})) = \mathbb{E}_{\boldsymbol{\theta}\sim\mathbb{U}(\boldsymbol{\theta})}[\log(\mathbb{U}(\boldsymbol{\theta})) - \log(\mathbb{V}(\boldsymbol{\theta}))] \tag{55}$$

$$= \frac{1}{2}\log\frac{|\boldsymbol{\Sigma}_p|}{|\boldsymbol{\Sigma}_s|} - \frac{1}{2}\mathbb{E}_{\boldsymbol{\theta}\sim\mathbb{U}(\boldsymbol{\theta})}(\boldsymbol{\theta} - \boldsymbol{\mu}_s)^T\boldsymbol{\Sigma}_s^{-1}(\boldsymbol{\theta} - \boldsymbol{\mu}_s) + \frac{1}{2}\mathbb{E}_{\boldsymbol{\theta}\sim\mathbb{U}(\boldsymbol{\theta})}(\boldsymbol{\theta} - \boldsymbol{\mu}_p)^T\boldsymbol{\Sigma}_p^{-1}(\boldsymbol{\theta} - \boldsymbol{\mu}_p) \tag{56}$$

$$= \frac{1}{2}[\log\frac{|\boldsymbol{\Sigma}_p|}{|\boldsymbol{\Sigma}_s|} - q + (\boldsymbol{\mu}_s - \boldsymbol{\mu}_p)^T\boldsymbol{\Sigma}_p^{-1}(\boldsymbol{\mu}_s - \boldsymbol{\mu}_p) + Tr(\boldsymbol{\Sigma}_p^{-1}\boldsymbol{\Sigma}_s)] \tag{57}$$

We assume the following inequality:

$$\log\frac{|\boldsymbol{\Sigma}_p|}{|\boldsymbol{\Sigma}_s|} + Tr(\boldsymbol{\Sigma}_p^{-1}\boldsymbol{\Sigma}_s) \le R + q, \quad R \ge 0 \tag{58}$$

Therefore,

$$D_{\mathrm{KL}}(\mathbb{U}(\boldsymbol{\theta})||\mathbb{V}(\boldsymbol{\theta})) \le \frac{1}{2}[R + (\boldsymbol{\mu}_s - \boldsymbol{\mu}_p)^T\boldsymbol{\Sigma}_p^{-1}(\boldsymbol{\mu}_s - \boldsymbol{\mu}_p)] \tag{59}$$

According to [50], we have the following identity:

For a random variable $\boldsymbol{\theta} \sim \mathcal{N}(\boldsymbol{\mu}, \boldsymbol{\Sigma})$

$$\mathbb{E}_{\boldsymbol{\theta}\sim\mathcal{N}(\boldsymbol{\mu},\boldsymbol{\Sigma})}(\boldsymbol{\theta} - \boldsymbol{\mu}')^T\boldsymbol{A}(\boldsymbol{\theta} - \boldsymbol{\mu}') = (\boldsymbol{\mu} - \boldsymbol{\mu}')^T\boldsymbol{A}(\boldsymbol{\mu} - \boldsymbol{\mu}') + Tr(\boldsymbol{A}\boldsymbol{\Sigma}) \tag{60}$$

where $Tr$ denotes the trace of A matrix. Therefore, according to Eq. (60), we have the following two equations 61 and 62.

$$\mathbb{E}_{\boldsymbol{\theta}\sim\mathbb{U}(\boldsymbol{\theta})}(\boldsymbol{\theta} - \boldsymbol{\mu}_s)^T\boldsymbol{\Sigma}_s^{-1}(\boldsymbol{\theta} - \boldsymbol{\mu}_s) = (\boldsymbol{\mu}_s - \boldsymbol{\mu}_s)^T\boldsymbol{\Sigma}_s^{-1}(\boldsymbol{\mu}_s - \boldsymbol{\mu}_s) + Tr(\boldsymbol{\Sigma}_s^{-1}\boldsymbol{\Sigma}_s) = q \tag{61}$$

$$\mathbb{E}_{\boldsymbol{\theta}\sim\mathbb{U}(\boldsymbol{\theta})}(\boldsymbol{\theta} - \boldsymbol{\mu}_p)^T\boldsymbol{\Sigma}_p^{-1}(\boldsymbol{\theta} - \boldsymbol{\mu}_p) = (\boldsymbol{\mu}_s - \boldsymbol{\mu}_p)^T\boldsymbol{\Sigma}_p^{-1}(\boldsymbol{\mu}_s - \boldsymbol{\mu}_p) + Tr(\boldsymbol{\Sigma}_p^{-1}\boldsymbol{\Sigma}_s) \tag{62}$$

We set $\boldsymbol{\gamma} = \boldsymbol{\Sigma}_p^{-\frac{1}{2}}(\boldsymbol{\mu}_s - \boldsymbol{\mu}_p)$. Then, $||\boldsymbol{\gamma}||^2 = (\boldsymbol{\mu}_s - \boldsymbol{\mu}_p)^T\boldsymbol{\Sigma}_p^{-1}(\boldsymbol{\mu}_s - \boldsymbol{\mu}_p)$.

If $\boldsymbol{\gamma} \sim N(\boldsymbol{0}, \tau^2\boldsymbol{I})$, according to [31], we have the following inequality with probability of $1 - \frac{1}{n}$

$$||\boldsymbol{\gamma}||^2 \le \tau^2(q + 2\sqrt{q\log n} + 2\log n) \le \tau^2(\sqrt{q} + \sqrt{2\log n})^2 \tag{63}$$

Then we partition the space of $\boldsymbol{\mu}_s$ into two disjoint area that satisfy $(\boldsymbol{\mu}_s - \boldsymbol{\mu}_p)^T\boldsymbol{\Sigma}_p^{-1}(\boldsymbol{\mu}_s - \boldsymbol{\mu}_p) \le 2\epsilon - R$ and $(\boldsymbol{\mu}_s - \boldsymbol{\mu}_p)^T\boldsymbol{\Sigma}_p^{-1}(\boldsymbol{\mu}_s - \boldsymbol{\mu}_p) > 2\epsilon - R$.

(1) In the case of $(\boldsymbol{\mu}_s - \boldsymbol{\mu}_p)^T\boldsymbol{\Sigma}_p^{-1}(\boldsymbol{\mu}_s - \boldsymbol{\mu}_p) \le 2\epsilon - R$, we take the maximum loss over $\boldsymbol{\mu}_s$, we have the following inequality:

$$\mathbb{E}_{\boldsymbol{\theta}\sim\mathbb{U}(\boldsymbol{\theta})}[\mathcal{L}_{\mathcal{T}}^{CL}(\boldsymbol{\theta})] \le \max_{(\boldsymbol{\mu}_s - \boldsymbol{\mu}_p)^T\boldsymbol{\Sigma}_p^{-1}(\boldsymbol{\mu}_s - \boldsymbol{\mu}_p)\le 2\epsilon - R}\mathbb{E}_{\boldsymbol{\theta}\sim\mathbb{U}(\boldsymbol{\theta})}\mathcal{L}^{CL}(\boldsymbol{\theta}) \tag{64}$$

(2) For the case of $(\boldsymbol{\mu}_s - \boldsymbol{\mu}_p)^T\boldsymbol{\Sigma}_p^{-1}(\boldsymbol{\mu}_s - \boldsymbol{\mu}_p) > 2\epsilon - R$, we have $\mathcal{L}_{\mathcal{T}}^{CL}(\boldsymbol{\theta}) \le C$

Combining case (1) and (2), we can obtain the following generalization bound:

$$D_{\mathrm{KL}}(\mathbb{U}, \mathbb{V}) \le \frac{1}{2}[(\boldsymbol{\mu}_s - \boldsymbol{\mu}_p)^T\boldsymbol{\Sigma}_p^{-1}(\boldsymbol{\mu}_s - \boldsymbol{\mu}_p) + R + q - q] \le \frac{1}{2}[||\boldsymbol{\gamma}||^2 + R] \tag{65}$$

$$\le \frac{1}{2}[\tau^2(\sqrt{q} + \sqrt{2\log n})^2 + R] \tag{66}$$

We have the following bound with probability of $1 - \frac{1}{n}$:

$$\mathbb{E}_{\boldsymbol{\theta}\sim\mathbb{U}(\boldsymbol{\theta})}[\mathcal{L}_{\mathcal{T}}^{CL}(\boldsymbol{\theta})] \leq (1-\frac{1}{n})\max_{(\boldsymbol{\mu}_s-\boldsymbol{\mu}_p)^T\boldsymbol{\Sigma}_p^{-1}(\boldsymbol{\mu}_s-\boldsymbol{\mu}_p)\leq 2\epsilon-R}\mathbb{E}_{\boldsymbol{\theta}\sim\mathbb{U}(\boldsymbol{\theta})}\mathcal{L}^{CL}(\boldsymbol{\theta}) + \frac{C}{n} \tag{67}$$

$$\leq (1-\frac{1}{n})\max_{D_{\mathrm{KL}}(\mathbb{U},\mathbb{V})\leq\epsilon}\mathbb{E}_{\boldsymbol{\theta}\sim\mathbb{U}(\boldsymbol{\theta})}\mathcal{L}^{CL}(\boldsymbol{\theta}) + \frac{C}{n} \tag{68}$$

Then, we can obtain the following generalization bound with probability of $1-\frac{1}{n}$:

$$\mathbb{E}_{\boldsymbol{\theta}\sim\mathcal{N}(\boldsymbol{\mu},\boldsymbol{\Sigma})}\sum_i^{i=T}\mathcal{L}_{\mathcal{D}_i}^{CL}(\boldsymbol{\theta}) \leq \max_{\mathbb{U}\in\mathcal{U}}\mathbb{E}_{\boldsymbol{\theta}\sim\mathbb{U}}\mathcal{L}^{CL}(\boldsymbol{\theta}) + \frac{C}{N_T+\zeta\sum_{i=1}^{i=T-1}N_i} + \tag{69}$$

$$\sqrt{\frac{\tau^2(\sqrt{q}+\sqrt{2\log(N_T+\zeta\sum_{i=1}^{i=T-1}N_i)})^2 + R + 2\log(\frac{N_T+\zeta\sum_{i=1}^{i=T-1}N_i}{\delta})}{4(N_T+\zeta\sum_{i=1}^{i=T-1}N_i-1)}}$$

$\square$

In this theorem, we provide the theoretical guarantee for the generalization analysis of our proposed method. This bound indicates by optimizing the MACL loss, our method tighten/reduce the generalization error of the CL method, thus improving the overall performance of our method.

## B  Equation Derivation

### B.1  Exponential Family Distribution Details

According to the definition of expectation, we can obtain the following equation:

$$\boldsymbol{\lambda}^1 := \mathbb{E}_{f(\boldsymbol{\theta};\boldsymbol{\mu},\boldsymbol{\Sigma})}\boldsymbol{\theta} = \boldsymbol{\mu} \tag{70}$$

According to the definition of covariance matrix,

$$\boldsymbol{\Sigma} := \mathbb{E}[(\boldsymbol{\theta}-\boldsymbol{\mu})(\boldsymbol{\theta}-\boldsymbol{\mu})^T] \tag{71}$$

$$= \mathbb{E}[\boldsymbol{\theta}\boldsymbol{\theta}^T - 2\boldsymbol{\mu}\boldsymbol{\theta} + \boldsymbol{\mu}\boldsymbol{\mu}^T] \tag{72}$$

$$= \mathbb{E}[\boldsymbol{\theta}\boldsymbol{\theta}^T] - \boldsymbol{\mu}\boldsymbol{\mu}^T \tag{73}$$

By rearranging the above equation, we can obtain the following:

$$\mathbb{E}[\boldsymbol{\theta}\boldsymbol{\theta}^T] = \boldsymbol{\mu}\boldsymbol{\mu}^T + \boldsymbol{\Sigma} \tag{74}$$

Then, the conclusion follows:

$$\boldsymbol{\lambda}^1 := \mathbb{E}_{f(\boldsymbol{\theta};\boldsymbol{\mu},\boldsymbol{\Sigma})}\boldsymbol{\theta} = \boldsymbol{\mu}, \quad \boldsymbol{\lambda}^2 := \mathbb{E}_{f(\boldsymbol{\theta};\boldsymbol{\mu},\boldsymbol{\Sigma})}\boldsymbol{\theta}\boldsymbol{\theta}^T = \boldsymbol{\mu}\boldsymbol{\mu}^T + \boldsymbol{\Sigma} \tag{75}$$

### B.2  Worst-Case Gaussian Distribution NGD Derivations

**Gradient of Loss $\mathcal{L}(\boldsymbol{\lambda})$ With Respect to $\boldsymbol{\lambda}$**  Taking gradient with respect to $\boldsymbol{\lambda}$ as the following:

$$\nabla_{\boldsymbol{\lambda}^1}\mathcal{L}(\boldsymbol{\lambda}) = \nabla_{\boldsymbol{\mu}}\mathcal{L}(\boldsymbol{\lambda})\frac{\partial\boldsymbol{\mu}}{\partial\boldsymbol{\lambda}^1} + \nabla_{\boldsymbol{\Sigma}}\mathcal{L}(\boldsymbol{\lambda})\frac{\partial\boldsymbol{\Sigma}}{\partial\boldsymbol{\lambda}^1} = \nabla_{\boldsymbol{\mu}}\mathcal{L}(\boldsymbol{\lambda}) - 2\nabla_{\boldsymbol{\Sigma}}\mathcal{L}(\boldsymbol{\lambda})\boldsymbol{\mu} \tag{76}$$

In Eq. (76), the second equality is because the identity: $\frac{\partial\boldsymbol{\mu}}{\partial\boldsymbol{\lambda}^1} = \mathbf{1}, \quad \frac{\partial\boldsymbol{\Sigma}}{\partial\boldsymbol{\lambda}^1} = \frac{\partial\boldsymbol{\Sigma}}{\partial\boldsymbol{\mu}} = -2\boldsymbol{\mu}$. (by Eq. (75))

$$\nabla_{\boldsymbol{\lambda}^2}\mathcal{L}(\boldsymbol{\lambda}) = \nabla_{\boldsymbol{\mu}}\mathcal{L}(\boldsymbol{\lambda})\frac{\partial\boldsymbol{\mu}}{\partial\boldsymbol{\lambda}^2} + \nabla_{\boldsymbol{\Sigma}}\mathcal{L}(\boldsymbol{\lambda})\frac{\partial\boldsymbol{\Sigma}}{\partial\boldsymbol{\lambda}^2} = \nabla_{\boldsymbol{\Sigma}}\mathcal{L}(\boldsymbol{\lambda}) \tag{77}$$

In Eq. (77), the second equality is because the identity: $\frac{\partial\boldsymbol{\mu}}{\partial\boldsymbol{\lambda}^2} = \mathbf{0}, \quad \frac{\partial\boldsymbol{\Sigma}}{\partial\boldsymbol{\lambda}^2} = \mathbf{1}$ (by Eq. (75))

According to Eq. (5), we set the natural parameters as:

$$\boldsymbol{\phi}^1 := \boldsymbol{\Sigma}^{-1}\boldsymbol{\mu}, \quad \boldsymbol{\phi}^2 := -\frac{1}{2}\boldsymbol{\Sigma}^{-1} \tag{78}$$

- (1) *NGD with respect to $\phi^2$*: According to Eq. (20 and 77), NGD with respect to $\phi^2$ can be obtained as:

$$-\frac{1}{2}\boldsymbol{\Sigma}_{i+1}^{-1} = -\frac{1}{2}\boldsymbol{\Sigma}_i^{-1} - \eta\nabla_{\boldsymbol{\lambda}^2}\mathcal{L}(\boldsymbol{\lambda}_i) = -\frac{1}{2}\boldsymbol{\Sigma}_i^{-1} - \eta\nabla_{\boldsymbol{\Sigma}}\mathcal{L}(\boldsymbol{\lambda}_i) \tag{79}$$

Then, obtain the following update:

$$\boldsymbol{\Sigma}_{i+1}^{-1} = \boldsymbol{\Sigma}_i^{-1} + 2\eta\nabla_{\boldsymbol{\Sigma}}\mathcal{L}(\boldsymbol{\lambda}_i) \tag{80}$$

- (2) *NGD with respect to $\phi^1$*: Similarly, according to Eq. (20 and 76), NGD with respect to $\phi^1$ can be obtained as:

$$\boldsymbol{\Sigma}_{i+1}^{-1}\boldsymbol{\mu}_{i+1} = \boldsymbol{\Sigma}_i^{-1}\boldsymbol{\mu}_i - \eta(\nabla_{\boldsymbol{\mu}}\mathcal{L}(\boldsymbol{\lambda}_i) - 2\nabla_{\boldsymbol{\Sigma}}\mathcal{L}(\boldsymbol{\lambda}_i)\boldsymbol{\mu}_i) \tag{81}$$

By simplifying and rearranging Eq. (81), the following update for $\boldsymbol{\mu}$:

$$\boldsymbol{\mu}_{i+1} = \boldsymbol{\mu}_i - \eta\boldsymbol{\Sigma}_{i+1}\nabla_{\boldsymbol{\mu}}\mathcal{L}(\boldsymbol{\lambda}_i) \tag{82}$$

**Mean and Covariance Updates Derivations**  Following the results in [56, 26], we can obtain the following equation:

$$\nabla_{\boldsymbol{\mu}}\mathbb{E}_{\boldsymbol{\theta}\sim u(\boldsymbol{\theta})}\mathcal{L}(\boldsymbol{\mu}, \boldsymbol{\Sigma}) = \mathbb{E}_{\boldsymbol{\theta}\sim u(\boldsymbol{\theta})}\nabla_{\boldsymbol{\theta}}\mathcal{L}(\boldsymbol{\mu}, \boldsymbol{\Sigma}) \tag{83}$$

$$\nabla_{\boldsymbol{\Sigma}}\mathbb{E}_{\boldsymbol{\theta}\sim u(\boldsymbol{\theta})}\mathcal{L}(\boldsymbol{\mu}, \boldsymbol{\Sigma}) = \frac{1}{2}\mathbb{E}_{\boldsymbol{\theta}\sim u(\boldsymbol{\theta})}\nabla_{\boldsymbol{\theta}\boldsymbol{\theta}}^2\mathcal{L}(\boldsymbol{\mu}, \boldsymbol{\Sigma}) \tag{84}$$

Then, we only need to calculate $\mathbb{E}_{\boldsymbol{\theta}\sim u(\boldsymbol{\theta})}\nabla_{\boldsymbol{\theta}}\mathcal{L}(\boldsymbol{\mu}, \boldsymbol{\Sigma})$ and $\mathbb{E}_{\boldsymbol{\theta}\sim u(\boldsymbol{\theta})}\nabla_{\boldsymbol{\theta}\boldsymbol{\theta}}^2\mathcal{L}(\boldsymbol{\mu}, \boldsymbol{\Sigma})$. Here, since we assumed a general CL Gaussian distribution for the current CL parameter distribution, i.e., $\mathbb{V}(\boldsymbol{\theta}) = \mathcal{N}(\boldsymbol{\theta}|\boldsymbol{\mu}_0, \boldsymbol{\Sigma}_0)$ and neighbourhood distribution, i.e., $\mathbb{U}(\boldsymbol{\theta}) = \mathcal{N}(\boldsymbol{\theta}|\boldsymbol{\mu}, \boldsymbol{\Sigma})$. The detailed derivations for the gradient are present in the following:

$$\nabla_{\boldsymbol{\mu}}\mathbb{E}_{\boldsymbol{\theta}\sim u(\boldsymbol{\theta})}\mathcal{L}(\boldsymbol{\mu}, \boldsymbol{\Sigma}) = -\mathbb{E}_{\boldsymbol{\theta}\sim u(\boldsymbol{\theta})}\nabla_{\boldsymbol{\theta}}\mathcal{L}^{CL}(\boldsymbol{\theta}) + \alpha\mathbb{E}_{\boldsymbol{\theta}\sim u(\boldsymbol{\theta})}[\nabla_{\boldsymbol{\theta}}\log u(\boldsymbol{\theta}) - \nabla_{\boldsymbol{\theta}}\log v(\boldsymbol{\theta})] \tag{85}$$

$$= -\mathbb{E}_{\boldsymbol{\theta}\sim u(\boldsymbol{\theta})}\nabla_{\boldsymbol{\theta}}\mathcal{L}^{CL}(\boldsymbol{\theta}) - \alpha\mathbb{E}_{\boldsymbol{\theta}\sim u(\boldsymbol{\theta})}(\boldsymbol{\theta} - \boldsymbol{\mu})\boldsymbol{\Sigma}^{-1} + \alpha\mathbb{E}_{\boldsymbol{\theta}\sim u(\boldsymbol{\theta})}(\boldsymbol{\theta} - \boldsymbol{\mu}_0)\boldsymbol{\Sigma}_0^{-1} \tag{86}$$

$$= \mathbb{E}_{\boldsymbol{\theta}\sim u(\boldsymbol{\theta})}[-\nabla_{\boldsymbol{\theta}}\mathcal{L}^{CL}(\boldsymbol{\theta}) + \alpha(\boldsymbol{\mu} - \boldsymbol{\mu}_0)\boldsymbol{\Sigma}_0^{-1}] \tag{87}$$

$$\nabla_{\boldsymbol{\Sigma}}\mathbb{E}_{\boldsymbol{\theta}\sim u(\boldsymbol{\theta})}\mathcal{L}(\boldsymbol{\mu}, \boldsymbol{\Sigma}) = -\frac{1}{2}\mathbb{E}_{\boldsymbol{\theta}\sim u(\boldsymbol{\theta})}\nabla_{\boldsymbol{\theta}\boldsymbol{\theta}}^2\mathcal{L}^{CL}(\boldsymbol{\theta}) + \alpha\mathbb{E}_{\boldsymbol{\theta}\sim u(\boldsymbol{\theta})}[\log u(\boldsymbol{\theta}) - \log v(\boldsymbol{\theta})] \tag{88}$$

$$= -\frac{1}{2}\mathbb{E}_{\boldsymbol{\theta}\sim u(\boldsymbol{\theta})}\nabla_{\boldsymbol{\theta}\boldsymbol{\theta}}^2\mathcal{L}^{CL}(\boldsymbol{\theta}) + \frac{\alpha}{2}\mathbb{E}_{\boldsymbol{\theta}\sim u(\boldsymbol{\theta})}[\nabla_{\boldsymbol{\theta}\boldsymbol{\theta}}^2\log u(\boldsymbol{\theta}) - \nabla_{\boldsymbol{\theta}\boldsymbol{\theta}}^2\log v(\boldsymbol{\theta})] \tag{89}$$

$$= -\frac{1}{2}\mathbb{E}_{\boldsymbol{\theta}\sim u(\boldsymbol{\theta})}\nabla_{\boldsymbol{\theta}\boldsymbol{\theta}}^2\mathcal{L}^{CL}(\boldsymbol{\theta}) + \frac{\alpha}{2}\mathbb{E}_{\boldsymbol{\theta}\sim u(\boldsymbol{\theta})}[-\boldsymbol{\Sigma}^{-1} + \boldsymbol{\Sigma}_0^{-1}] \tag{90}$$

$$= \frac{1}{2}\mathbb{E}_{\boldsymbol{\theta}\sim u(\boldsymbol{\theta})}[-\nabla_{\boldsymbol{\theta}\boldsymbol{\theta}}^2\mathcal{L}^{CL}(\boldsymbol{\theta}) - \alpha\boldsymbol{\Sigma}^{-1} + \alpha\boldsymbol{\Sigma}_0^{-1}] \tag{91}$$

Plug-in the gradient derivation into Eq. (82 and 80), we can obtain the following results:

$$\boldsymbol{\Sigma}_{i+1}^{-1} = \boldsymbol{\Sigma}_i^{-1} + \eta\mathbb{E}_{\boldsymbol{\theta}\sim u(\boldsymbol{\theta})}[-\nabla_{\boldsymbol{\theta}\boldsymbol{\theta}}^2\mathcal{L}^{CL}(\boldsymbol{\theta}) - \alpha\boldsymbol{\Sigma}_i^{-1} + \alpha\boldsymbol{\Sigma}_0^{-1}] \tag{92}$$

$$= (1 - \eta\alpha)\boldsymbol{\Sigma}_i^{-1} + \eta\mathbb{E}_{\boldsymbol{\theta}\sim u(\boldsymbol{\theta})}[-\nabla_{\boldsymbol{\theta}\boldsymbol{\theta}}^2\mathcal{L}^{CL}(\boldsymbol{\theta}) + \alpha\boldsymbol{\Sigma}_0^{-1}] \tag{93}$$

$$\boldsymbol{\mu}_{i+1} = \boldsymbol{\mu}_i - \eta\boldsymbol{\Sigma}_{i+1}\mathbb{E}_{\boldsymbol{\theta}\sim u(\boldsymbol{\theta})}[-\nabla_{\boldsymbol{\theta}}\mathcal{L}^{CL}(\boldsymbol{\theta}) + \alpha(\boldsymbol{\mu}_i - \boldsymbol{\mu}_0)\boldsymbol{\Sigma}_0^{-1}] \tag{94}$$

Finally, by using single sample from distribution $\mathbb{U}$ with density $\boldsymbol{\theta} \sim u(\boldsymbol{\theta})$ to approximate the expectation. By plug-in $\boldsymbol{\Sigma} = \mathrm{diag}(\boldsymbol{\sigma}^2)$ and $\boldsymbol{\Sigma}_0 = \mathrm{diag}(\boldsymbol{\rho}^2)$ into the above equations, we can obtain the following updates:

$$\boldsymbol{\mu}_{i+1} = \boldsymbol{\mu}_i + \eta\boldsymbol{\sigma}_{i+1}^2[\nabla_{\boldsymbol{\theta}}\mathcal{L}^{CL}(\boldsymbol{\theta}_i) - \alpha(\boldsymbol{\mu}_i - \boldsymbol{\mu}_0)\boldsymbol{\rho}^{-2}] \tag{95}$$

$$\boldsymbol{\sigma}_{i+1}^{-2} = (1 - \eta\alpha)\boldsymbol{\sigma}_i^{-2} + \eta[-\nabla_{\boldsymbol{\theta}\boldsymbol{\theta}}^2\mathcal{L}^{CL}(\boldsymbol{\theta}_i) + \alpha\boldsymbol{\rho}^{-2}] \tag{96}$$

# C   Baseline Details

- **EWC** [28]: EWC endeavors to alleviate forgetting in continual learning through the utilization of a weighted weight regularization technique based on the Fisher information matrix.

- **CPR** [10]: Drawing on neural networks with wide local minima and principles from information theory, CPR introduces an extra regularization term. This term aims to maximize the entropy of a classifier's output probabilities, thereby reaching wider local minima to enhance generalization.

- **GPM** [60]: A CL model acquires new skills by adjusting its parameters through gradient steps that move orthogonal to the gradient subspaces considered vital for previous tasks. The Gradient Projection Memory (GPM) establishes these subspaces by analyzing network activations following the completion of each task using Singular Value Decomposition (SVD), then preserves them in memory.

- **HAT** [63]: HAT is a task-driven hard attention mechanism that retains information from prior tasks while ensuring it doesn't interfere with the current task's learning process.

- **A-GEM** [14]: AGEM aims to guarantee that, at every training step, the average loss of episodic memory over past tasks does not rise, thus mitigating the risk of forgetting previously acquired knowledge.

- **Gradient-Based Sample Selection (GSS-Greedy)** [2]: The goal is to populate the memory buffer with a diverse set of examples, using the data gradient as a feature for sample selection. For comparison, we opt for the efficient GSS-Greedy version.

- **ER** [15]: This method stores a subset of examples from previous tasks using reservoir sampling [15]. During each iteration, we randomly replay a subset of examples from the memory buffer.

- **DER++** [7]: This method combines experience replay with knowledge distillation to further improve the effectiveness of experience replay.

- **ER-ACE** [8]: They discovered that ER causes significant overlap between the representations of newly added classes and previous ones, resulting in highly disruptive parameter updates. From this empirical analysis, they proposed a new method to address this issue by protecting the learned representations from drastic adaptations when incorporating new classes. Their approach uses an asymmetric update rule that pushes new classes to adapt to the older ones, rather than the reverse. This technique is particularly effective at task boundaries, where much of the forgetting typically happens.

- **LODE** [36]: They conducted an in-depth analysis of the impacts of distinguishing between new and old classes, as well as among new classes, finding that these two learning objectives result in varying degrees of forgetting. Consequently, combining these objectives negatively affects the performance of the CL model. To address this, LODE separates the two objectives for new tasks by decoupling the loss associated with them. This approach allows LODE to assign different weights to each objective, leading to better performance compared to methods that use a coupled loss.

# D   More Experimental Results

## D.1   Hyperparameter Sensitivity Analysis

Table 5: Analysis of hyperparameter $\eta$ on CIFAR100 and Tiny-ImageNet in the setting of task-IL.

| $\eta$ | 0.0 | 1e-7 | 1e-6 | 1e-5 | 3×1e-5 |
|---|---|---|---|---|---|
| CIFAR100 | 75.64±0.60 | 77.16±0.42 | 77.69±0.37 | 77.53±0.89 | 76.97±0.46 |
| Tiny-ImageNet | 51.91±0.68 | 53.12±0.82 | 54.03±0.79 | 54.46±0.91 | 51.62±0.55 |

## D.2 Benefit of NGD

Table 6: Benefit of MACL-NGD vs. MACL-GD on CIFAR100 and Tiny-ImageNet in the setting of task-IL.

| method | DER++ | DER++(MACL-NGD) | DER++(MACL-GD) |
|---|---|---|---|
| CIFAR100 | 75.64±0.60 | 77.53±0.89 | 76.62±0.53 |
| Tiny-ImageNet | 51.91±0.68 | 54.03±0.79 | 52.97±0.71 |

## D.3 Online CL results

Table 7: Online CL Results on CIFAR100 under the blurry boundary setting

| Memory Size | 1000 | 2000 | 5000 |
|---|---|---|---|
| MKD(PCR) | $35.6 \pm 0.66$ | $44.95 \pm 0.42$ | $54.87 \pm 0.39$ |
| MKD(PCR) + MACL | $\mathbf{37.2 \pm 0.53}$ | $\mathbf{46.17 \pm 0.51}$ | $\mathbf{56.21 \pm 0.43}$ |

Table 8: Online CL Results on Tiny-ImageNet under the blurry boundary setting

| Memory Size | 2000 | 5000 | 10000 |
|---|---|---|---|
| MKD(PCR) | $17.33 \pm 1.28$ | $29.58 \pm 0.60$ | $38.02 \pm 1.64$ |
| MKD(PCR) + MACL | $\mathbf{18.21 \pm 1.32}$ | $\mathbf{30.69 \pm 0.71}$ | $\mathbf{38.73 \pm 1.56}$ |

## D.4 Prompt-based CL results

We conducted an experiment integrating MACL with the SOTA prompt-based CL method, CODA-Prompt [65]. Our method operates on the parameters of prompt components and corresponding keys/attention vectors.

Table 9: CODA Prompt Results on ImageNet-R

| Number of Tasks | 10 | 20 |
|---|---|---|
| CODA-P | $75.45 \pm 0.56$ | $72.37 \pm 1.19$ |
| CODA-P + MACL | $\mathbf{76.39 \pm 0.67}$ | $\mathbf{73.42 \pm 1.23}$ |

## D.5 5-datasets results

Table 10: Comparison of methods on Class-IL and Task-IL on 5-datasets.

| Method | Class-IL | Task-IL |
|---|---|---|
| ER | $66.03 \pm 1.37$ | $92.58 \pm 1.26$ |
| ER+MACL | $\mathbf{67.32 \pm 1.18}$ | $\mathbf{93.21 \pm 1.08}$ |
| DER++ | $85.92 \pm 0.33$ | $87.16 \pm 0.21$ |
| DER++MACL | $\mathbf{87.23 \pm 0.51}$ | $\mathbf{87.51 \pm 0.30}$ |

## D.6 Effect of Different Architectures

Table 11: Overall accuracy with ResNet32 using a memory buffer of 2000 by integrating with MEMO.

| | MEMO | MEMO+MACL |
|---|---|---|
| accuracy | 58.49 | 59.61 |

Table 12: Overall accuracy with ViT using a memory buffer of 500 by integrating DER++ with MACL.

| | Class-IL | Task-IL |
|---|---|---|
| DER++ | $76.21 \pm 0.67$ | $96.72 \pm 0.31$ |
| DER++ MACL | $77.83 \pm 0.80$ | $97.31 \pm 0.46$ |

### D.7 ImageNet-R and CUB200 results

We conducted experiment on the recent CL datasets of ImageNet-R and CUB200 with pre-trained Vision Transformer (ViT), i.e., vit-base-patch16-224 as the backbone following the codebase of DER++. The results (memory size of 500) are shown in the following table.

Table 13: ImageNet-R Results

| Method | Class-IL | Task-IL |
|---|---|---|
| DER++ | $58.29 \pm 1.78$ | $86.93 \pm 0.32$ |
| DER++MACL | $\mathbf{60.51 \pm 1.65}$ | $\mathbf{87.56 \pm 0.41}$ |
| LODE | $74.98 \pm 0.21$ | $90.22 \pm 0.39$ |
| LODE+MACL | $\mathbf{75.51 \pm 0.26}$ | $\mathbf{90.81 \pm 0.28}$ |

Table 14: CUB200 Results

| Method | Class-IL | Task-IL |
|---|---|---|
| DER++ | $41.81 \pm 1.69$ | $87.16 \pm 1.09$ |
| DER++MACL | $\mathbf{43.07 \pm 1.53}$ | $\mathbf{88.03 \pm 0.97}$ |
| LODE | $66.87 \pm 0.35$ | $93.12 \pm 0.56$ |
| LODE+MACL | $\mathbf{67.53 \pm 0.51}$ | $\mathbf{93.42 \pm 0.37}$ |

### D.8 Efficiency Evaluation

Table 15: Running efficiency of MACL on CIFAR100 by training for a single epoch on CIFAR100.

| CL method | w/o MACL | w/ MACL |
|---|---|---|
| DER++ | 8.7 | 13.5 |
| ER-ACE | 6.3 | 10.2 |
| LODE | 13.2 | 20.8 |

## E Experiment Setup

### E.1 Dataset Statistics

Table 16: Dataset Statistics

| Dataset | Seq-CIFAR10 | Seq-CIFAR100 | Seq-TinyImageNet |
|---|---|---|---|
| Number of Tasks | 5 | 10 | 10 |
| Number of Classes | 10 | 100 | 200 |
| Number of Training Samples | 50,000 | 50,000 | 100,000 |
| Number of Test Samples | 10,000 | 10,000 | 10,000 |

