# OpenReview forum: "Model Sensitivity Aware Continual Learning"
_NeurIPS.cc/2024/Conference — NeurIPS 2024 poster_

### Official Review · Reviewer_g3Gj · 2024-07-03

**Soundness:** 3
**Presentation:** 3
**Contribution:** 2
**Rating:** 6
**Confidence:** 4

**Summary:**

The paper "Model Sensitivity Aware Continual Learning" introduces a novel approach to continual learning (CL) that addresses the trade-off between retaining previously acquired knowledge and excelling in new task performance. Traditional CL models often face the dilemma of catastrophic forgetting or overfitting to new tasks. The proposed method mitigates model parameter sensitivity to updates, ensuring that minor parameter changes do not significantly impact the model's performance. This is achieved by optimizing the model's performance based on the worst-case scenario within a parameter distribution neighborhood. The approach is compatible with existing CL methodologies, offering substantial improvements in retaining old knowledge and enhancing new task performance, supported by theoretical and empirical evidence.

**Strengths:**

Originality: The concept of reducing model parameter sensitivity to improve both retention and new task performance is innovative and orthogonal to existing methods.
Quality: The methodological rigor and comprehensive theoretical analysis are commendable.
Clarity: The paper is well-structured, with clear and precise explanations of the proposed method and its advantages.
Significance: The ability to improve CL models' performance without sacrificing old knowledge retention is a crucial advancement in the field.

**Weaknesses:**

The computational complexity of the proposed method, particularly the calculation of the Fisher Information Matrix, might be a concern for large-scale applications.
While the method is demonstrated to be effective in offline CL scenarios, its applicability to online CL is not explored.
The hyperparameter sensitivity, especially regarding the learning rate η, requires careful tuning, which might limit the method's usability in different settings without additional hyperparameter optimization.
The datasets used in this paper are outdated and could consider more recent datasets.

**Questions:**

Can the authors provide more insights into the computational overhead introduced by the proposed method, particularly in large-scale datasets or real-time applications?
Have the authors considered any strategies for optimizing hyperparameters in a more automated fashion to improve the method's usability across different datasets and tasks?
Are there any plans to extend this approach to online continual learning scenarios? If so, what challenges do the authors anticipate?

**Limitations:**

The authors have adequately addressed the limitations of their work, particularly in the context of offline CL scenarios. However, exploring the method's applicability to online CL could provide a more comprehensive understanding of its potential. Additionally, a more detailed discussion on the computational complexity and strategies to mitigate it would be beneficial.

---

> ### Author Rebuttal · Authors · 2024-08-06
>
> We sincerely appreciate your valuable suggestions and would like to express our gratitude.
>
> **Q1**  computational complexity of the proposed method and Fisher Information Matrix (FIM) for large-scale applications.
>
> **A1** Thank you for your question! Although the FIM calculation is costly, our method circumvent this difficulty by calculating the FIM in the dual parameter space of exponential family distribution, i.e., the expectation parameter space. According to theorem 3.1, the FIM in the natural parameter space is equal to the *gradient in its dual space, i.e., the expectation parameter space*. Therefore, the computation cost of FIM is not high. In addition, we conducted experiments to compare the running efficiency by integrating our proposed method (MACL) with multiple CL approaches, including DER++, ER-ACE, and LODE. The training time (in minutes) for a single epoch on ImageNet-R with ViT as backbone is shown below. Our method only marginally increases the training time by approximately 56% to 64%. This indicates that our method only introduces small computation cost and is efficient in practice.
>
> **running time on ImageNet-R (minutes)**
> | Method | w/o MACL | w/ MACL |
> | ------ | -------- | ------ |
> | DER++  | 1.52      | 2.46   |
> | ER-ACE | 1.16      | 1.81 |
> | LODE   | 1.98   |   3.25  |
> **Q2** Application in online CL
>
> **A2** Online Continual Learning (CL) focuses on learning from a data stream with unclear task boundaries. This poses the challenge of not being able to utilize task identity to calculate the FIM. Additionally, we need to maintain  computational efficiency. To address these challenges, we explore calculating the FIM in an online CL setting using a moving average approach. Specifically, we approximate the diagonal Hessian matrix at each time step using the gradient outer product, i.e., $H_t \approx g_tg_t^T$. To improve training efficiency, we calculate the FIM only in the first two residual network blocks (closer to the inputs) of ResNet (which consists of four residual network blocks). This is based on existing studies that show earlier layers, which calculate the input data representations, are more sensitive to task drift. We invite you to refer to the **global response Q1 and A1** for the results in online CL setting.
>
> **Q3**  hyperparameter optimization.
>
> **A3**   Thank you for your suggestions! We have explored automatic hyperparameter optimization to enhance the method's usability by utilizing the SMAC3 framework [1], a sample-efficient and automatic Bayesian optimization hyperparameter search strategy. The primary goal of Bayesian optimization for hyperparameter search is to optimize a black-box function. In the context of CL, the objective function is the average validation accuracy on the first three tasks, following [4].
> We set the hyperparameters to be optimized, specifically the learning rate $\eta$, within the range of $[1e-6, 1e-2]$. SMAC3 then performs the hyperparameter optimization process, iteratively suggesting new configurations to evaluate and updating its model based on the results. The selection of which hyperparameter configuration to evaluate next is determined by an acquisition function, which balances exploration (trying new configurations) and exploitation (focusing on configurations that are known to perform well). Initially, we perform hyperparameter optimization on the first three tasks, then use these optimized hyperparameters for subsequent tasks, following existing CL works [4]. The performance of the searched optimal hyperparameters on TinyImageNet with a memory buffer size of 500 is shown in the following table.
>
> **Hyperparameter Optimization for $\eta$ on Tiny-ImageNet**
>
> |  | Class-IL | Task-IL |
> | -------- | -------- | -------- |
> |  Manually Selected    |  20.17 $\pm$ 1.56    |   54.03 $\pm$ 0.79   |
> |  Automatically Selected    |  20.39 $\pm$ 1.32    |   54.27 $\pm$ 0.83   |
>
> **Q4**  Datasets outdated
>
> **A4** Thank you pointing out this! We conducted experiment on the recent CL datasets of ImageNet-R [2] and CUB200 [3] with pre-trained Vision Transformer (ViT), i.e., vit-base-patch16-224 as the backbone following the codebase of DER++. The results (memory size of 500) are shown in the following table.
>
>  **ImageNet-R Results**
>  | Method | Class-IL | Task-IL |
>  | -------- | -------- | -------- |
>  | DER++     |  58.29 $\pm$ 1.78  |     86.93 $\pm$ 0.32  |
>  | DER++MACL       | **60.51 $\pm$ 1.65**   |  **87.56 $\pm$ 0.41**      |
>  | LODE       | 74.98 $\pm$ 0.21     |   90.22 $\pm$ 0.39    |
>  | LODE+MACL       | **75.51 $\pm$ 0.26**    | **90.81 $\pm$ 0.28**    |
>
>  **CUB200 Results**
>  | Method | Class-IL | Task-IL |
>  | -------- | -------- | -------- |
>  | DER++     | 41.81 $\pm$ 1.69    |  87.16 $\pm$ 1.09    |
>  | DER++MACL       | **43.07 $\pm$ 1.53**  | **88.03 $\pm$ 0.97**        |
>  | LODE       |  66.87 $\pm$ 0.35    | 93.12 $\pm$ 0.56     |
>  | LODE+MACL       |  **67.53 $\pm$ 0.51**     |    **93.42 $\pm$ 0.37**   |
>
> **Q5** Can the authors provide more insights into the computational overhead introduced by the proposed method, particularly in large-scale datasets or real-time applications? Have the authors considered any strategies for optimizing hyperparameters in a more automated fashion to improve the method's usability across different datasets and tasks? Are there any plans to extend this approach to online continual learning scenarios? If so, what challenges do the authors anticipate?
>
> **A5**  Thank you for your questions! We invite you to refer to the question and answer in **Q1 to Q4** and  **A1 to A4**.
>
> Reference:
>
> [1] SMAC3: A Versatile Bayesian Optimization Package for Hyperparameter Optimization, JMLR 2022
>
> [2] The many faces of robustness: A critical analysis of out-of-distribution generalization. ICCV 2021
>
> [3] The caltech-ucsd birds-200-2011 dataset, 2011
>
> [4] Efficient Lifelong Learning with A-GEM, ICLR 2019

---

> > ### Comment · Reviewer_g3Gj · 2024-08-09
> > **Response to Rebuttal by Authors**
> >
> > Thanks for the detailed answers to my questions. I am impressed by the additional complexity analysis, parameter analysis, and more recent data experiments. I will be happy to see them in your future paper. For now, I will keep my score.

---

> ### Author Response · Authors · 2024-08-09
> **Thank you!**
>
> Thank you for your feedback!  We really appreciate your support!

---

### Official Review · Reviewer_kb2F · 2024-07-11

**Soundness:** 3
**Presentation:** 2
**Contribution:** 3
**Rating:** 6
**Confidence:** 3

**Summary:**

The paper presents a new method to address catastrophic forgetting and improve the learning ability of a new task in continual learning. They attribute these two objectives to parameter sensitivity. To address this problem, they propose to minimize the performance on the worst-case CL parameter distribution within the neighborhood of the current CL model. The approach is integrated with existing CL methods and evaluated on three continual learning benchmarks CIFAR10, CIFAR100, and Tiny-ImageNet.

**Strengths:**

Strength:

- To the best of my knowledge, it is a novel perspective to address the continual learning problem.
- Integrating the approach with current CL methods improves performance.
- The method is evaluated on two scenarios: task incremental learning and the more challenging one, class incremental learning.
- Multiple CL metrics are evaluated and analysis is provided.

**Weaknesses:**

- It is not clear whether the assumption of CL model parameters following a normal distribution could be valid for all models.

- I found it hard to easily understand the relation between the parameter sensitivity and the worst-case CL model.

- The presentation could be improved. Some sections are wordy. Those can be concise and some of the theoretical proofs in the appendix can be moved to the main paper to easily follow.

- I would expect to see forgetting reported in Table 3.

- Most baselines are a bit old.

**Questions:**

- It seems that you mostly focus on integrating the method with replay-based approaches. Is there a motivation behind it?

- Do the same findings generalize to a long sequence of tasks?

- From Table 1 and Table 2, it seems that the methods improve new task learning more than reducing catastrophic forgetting. Do you have some thoughts on that?

**Limitations:**

Yes.

---

> ### Author Rebuttal · Authors · 2024-08-06
>
> We deeply appreciate the thoughtfulness of your feedback and support!
>
> **Q1**  whether the assumption of CL model parameters following a normal distribution could be valid for all models.
>
> **A1** Thank you for your question!
>
> * The assumption that model parameters follow a Gaussian distribution is intended to balance good performance with computational efficiency. This assumption is widely adopted in Bayesian inference for neural network parameters because of its simplicity and scalability. The Gaussian distribution is computationally efficient to work with and can be easily integrated into various optimization and inference algorithms. This balance between performance and efficiency makes the Gaussian distribution a practical choice for modeling neural network parameters, particularly in complex tasks such as continual learning.
>
> * We extend the standard normal distribution of current CL model parameters into a more general Gaussian distribution update. Here, we assume a more general Gaussian distribution to model current CL parameter distribution, $V(\theta) = \mathcal{N}(\theta|\mu_0, \Sigma_0)$, where $\mu_0$ and $\Sigma_0$ denote the mean and covariance matrix of current CL model parameters, respectively.  We can obtain the following update equation:
>
>    $\Sigma^{-1}_{i+1}= (1 - \eta)\Sigma^{-1}_i + \eta \mathbb{E} _{\theta\sim u(\theta)} [-\nabla _{\theta\theta}^2 \mathcal{L} ^{CL}(\theta) + \Sigma_0^{-1}]$
>
>    $\mu_{i+1} = (1 - \eta)\mu_{i} + \eta \Sigma _{i+1} \mathbb{E} _{\theta \sim u(\theta)} [\nabla _{\theta} \mathcal{L}^{CL}(\theta)+ \mu_0 \Sigma _0^{-1}]$
>
>     Compared to standard normal distribution modeling assumption. Here, the above more general Gaussian parameter distribution incorporates one additional term of $\mu_0 \Sigma_0^{-1}$ and $\Sigma_0^{-1}$ for $\mu$ and $\Sigma$ update process, respectively. This allows for greater flexibility and adaptability in the update process.
>
> * Our proposed framework is both flexible and extendible. It can be adapted to handle more general and complex parameter distributions to capture a wider range of parameter variability. This flexibility enhances its applicability to more complex distributions.
>
> **Q2**  I found it hard to easily understand the relation between the parameter sensitivity and the worst-case CL model.
>
> **A2** Parameter sensitivity implies that small changes in parameters can lead to significant drops in performance. By focusing on the worst-case scenario, we are minimizing the potential performance degradation that can occur due to parameter updates. The model is designed to be more stable against the worst possible updates in parameters. If a model can perform well under the worst-case conditions, it will naturally perform well under less extreme conditions.
>
> This approach ensures that the model does not rely on any specific set of parameters to perform well, but rather maintains good performance across a wide range of possible parameter values to minimize performance degradation.  By preparing for the worst-case, the model minimizes the variability in its performance. This means that small changes in parameters are less likely to cause significant deviations in performance, thereby reducing parameter sensitivity.
>
> **Q3**  presentation could be improved.
>
> **A3** Thank you for your suggestions! We followed your advice and revised presentation in the revision.
>
> **Q4**  I would expect to see forgetting reported in Table 3.
>
> **A4**
> |   | **CIFAR-100 Class-IL** | **CIFAR-100 Task-IL** | **Tiny-ImageNet Class-IL** | **Tiny-ImageNet Task-IL** |
> |---------------|------------------------|-----------------------|----------------------------|--------------------------|
> | **ER**        | -51.29 ± 0.43 | -8.17 ± 0.25  | -63.76 ± 0.38 | **-17.12 ± 0.26** |
> | **ER+MACL**   | **-50.71 ± 0.58** | **-7.28 ± 0.19** | **-62.53 ± 0.46** | -17.59 ± 0.33 |
> | **DER++**     | -34.26 ± 0.31 | -7.83 ± 0.29  | -43.05 ± 0.42 | -14.56 ± 0.78 |
> | **DER++ + MACL** | **-33.32 ± 0.39** | **-7.56 ± 0.31** | **-42.27 ± 0.53** | **-13.82 ± 0.65** |
> | **LODE**      | **-21.15 ± 1.06** | -2.72 ± 1.32  | -37.32 ± 1.25 | -9.19 ± 1.07 |
> | **LODE + MACL** | -21.80 ± 1.23 | **-1.93 ± 1.17** | **-36.38 ± 1.03** | **-8.02 ± 1.68** |
>
> **Q5** Most baselines are a bit old.
>
> **A5**  Thank you for pointing out this! We added one SOTA class-incremental learning (CIL) MRFA [1], one SOTA online CL baseline [2] and prompt-based baseline [3]. We invite you to refer to the **global response Q1, Q3, Q4** and **A1, A3, A4**
>
> **Q6** motivation for focus on integrating the method with replay-based approaches.
>
> **A6** Thank you for your question! This is because memory-replay-based approaches usually achieves better performance and are more widely studied than other continual learning methods.
>
> **Q7** generalize to a long sequence of tasks?
>
> **A7** Thank you for your question, we invite you to refer to the **global response Q2** and **A2**.
>
> **Q8** From Table 1 and Table 2, it seems that the methods improve new task learning more than reducing catastrophic forgetting.
>
> **A8** Thank you for your  question! We believe that average accuracy is more easily improved because it benefits directly from the effective learning of new tasks and preservation of old task knowledge, which is often the primary focus of many continual learning methods. In contrast, reducing forgetting and improving backward transfer requires maintaining the performance of all previously learned tasks, which is a more complex and challenging problem.
>
> Reference:
>
> [1] Multi-layer Rehearsal Feature Augmentation for Class-Incremental Learning, ICML 2024
>
> [2] Rethinking Momentum Knowledge Distillation in Online Continual Learning, ICML 2024
>
> [3] CODA-Prompt: COntinual Decomposed Attention-based Prompting for Rehearsal-Free Continual Learning, CVPR 2023

---

> > ### Comment · Reviewer_kb2F · 2024-08-10
> > **Official Comment by Reviewer kb2F**
> >
> > Thank you for addressing my concerns and performing additional experiments. I will keep my positive score.

---

> > > ### Author Response · Authors · 2024-08-10
> > > **Thank you!**
> > >
> > > Thank you for your response! We sincerely appreciate your support!

---

### Official Review · Reviewer_YRBc · 2024-07-12

**Soundness:** 3
**Presentation:** 2
**Contribution:** 3
**Rating:** 5
**Confidence:** 5

**Summary:**

This paper presents a min-max optimization framework targeting at the model sensitivity for continual learning, where the authors claim that it can mitigate the abrupt change of the model parameters and thus simultaneously alleviates the problem of catastrophic forgetting of the past knowledge and overfitting to the current task. By assuming the posterior distribution of the model parameters as multivariate gaussian, it achieves effective min-max optimization. Some experimental results are are presented to demonstrate the model's better performance.

**Strengths:**

1. The motivation of mitigating the catastrophic forgetting by addressing the model sensitivity is promising and there exists some work showing this point.
2. The proof of the main claims in the paper are paired with detailed derivation, which provides additional source of reference for the readers.

**Weaknesses:**

1. The paper starts with treating the current continual learning objective as a whole without discussing the interplays among the different tasks or the data scarcity problem of the previous tasks, which is not a usual case. The oversimplification of assuming a $\mathcal{L}^{\text{CL}}(\theta)$ can be problematic in the sense that we generally want to treat different tasks differently (past vs. current) so that old knowledge retention and new knowledge acquisition can be achieved at the same time. To further illustrate my point, consider some other arbitrary learning scenarios, this "model sensitivity awareness" can be seamlessly applied. So please elaborate on why CL specifically benefits from this algorithm.
2. It is not clear what the generalization bound provided by theorem 4.3 implies. It still contains the proxy training target for continual learning $\mathcal{L}^{\text{CL}}$ and seems not very informative about the CL algorithm.
3. The organization and presentation of the paper is not confusing, especially for Section 3.2. There are theorems, together with too detailed derivation of the intermediate results, I would suggest the author re-organize the material and only summarize the most important results in the main paper, and put the not-so-relevant content in the appendix.
4. The experimental results seem not rigorously verified. As far as I know, the CIL results on CIFAR100 dataset should start with 40-50% overall accuracy but not this low in the paper (too many references and I will just skip providing them). Please provide more demonstration on this.

**Questions:**

1. I find the formulation of the fundamental optimization problem in this work confusing. In Eq. 7,
$$
\min_\theta\max_{U\in\mathcal{U}} E_{\theta\sim U(\theta)}[\mathcal{L}^{\text{CL}}(\theta)],
$$
$$
\text{s.t.}\quad \mathcal{U} = \\{U: D_{\text{KL}}(U, V) < \epsilon \\},
$$
the outer minimization over $\theta$ is not taking any effect in the inner maximization as it is integrated out over $U$. I assume this $V$ should be dependent on $\theta$? But in the main paper, the authors claim $V=\mathcal{N}(0, I)$. Please elaborate on this.

**Limitations:**

Yes. The authors addressed the limitations in Section 6.

---

> ### Author Rebuttal · Authors · 2024-08-06
>
> We would like to express our sincere gratitude for your valuable comments.
>
> **Q1** treating current continual learning objective as a whole without discussing the interplays among the different tasks or the data scarcity problem of the previous tasks.  why CL specifically benefits from this algorithm.
>
>  **A1**  Thanks for your question!
>
>  * We would like to clarify that our notation of $L^{CL}(\theta)$ does not explicitly treat the continual leanring process as a whole. Instead, this is an abstract and unified loss notation that can represent the memory-based loss, regularization-based loss, etc. It simplifies the derivation process and guarantees the general applicability of our proposed method. Specifically, $L^{CL}(\theta) = L_{CE}(\theta) + L_{f}(\theta)$. Here, the new task loss is $L_{new} = L_{CE}(\theta)$ and the old task loss approximation is $L_{old} = L_{f}(\theta)$. Thus, it can be further formulated as $L^{CL}(\theta) = L_{new}(\theta) + L_{old}(\theta)$, which describes the interplay between new and old tasks.
>
> * The data scarcity problem of the previous tasks is incorporated into the loss of $L_{old}(\theta)$. For example, if adopting a memory-replay-based method to mitigate forgetting, $L_{old}(\theta) = L_{(x,y) \sim M}(x, y, \theta)$, i.e., the memory-replay loss, where $M$ is a memory buffer that stores a small number of data from previous tasks. If adopting a regularization-based approach, $L_{old}(\theta) $ is a regularization term to mitigate the parameter drift from previous task parameters.
>
> * Our "model-sensitivity-aware" loss operates on both $L_{old}(\theta)$ and $L_{new}(\theta)$ at the same time. Operating on $L_{old}(\theta)$ reduces the parameter sensitivity on old tasks to mitigate forgetting on old tasks. Operating on $L_{new}(\theta)$ reduces the parameter sensitivity on new tasks to reduce overfitting on the new tasks, thus leading to better performance on new tasks. Therefore, our method mitigates the trade-off between old task knowlege retention and new task learning by improving the performance on new and old tasks simultaneously.
>
> **Q2** what the generalization bound in theorem 4.3 implies.
>
> **A2**
> * **Generalization bound**:
>     * For the first term on the right-hand side (RHS) of the generalization bound, in traditional CL, the training loss does not bound the generalization error on unseen test data. In contrast, by adopting our model sensitivity-aware CL approach, the model sensitivity-aware training loss (the first term) plus an additional complexity term (the third term) can bound the generalization error on unseen test data. Therefore, this implies that by minimizing the model sensitivity-aware loss, i.e., $\max _{\mathbb{U} \in \mathcal{U}} \mathbb{E} _{\theta \sim \mathbb{U}} L ^{CL} _{T}(\theta)$, we can indeed reduce the generalization error.
>     * The third term in the generalization bound is the model complexity term, i.e., $\sqrt{\frac{\tau^2(\sqrt{q} + \sqrt{2\log n})^2 + R + 2\log(\frac{n}{\delta})}{4(n-1)}}$, represented by the KL divergence between the posterior distribution 𝑄 and the prior distribution 𝑃 and simplifid to the current form. This term quantifies the complexity of the posterior distribution relative to the prior. A higher KL divergence indicates that the posterior distribution 𝑄 deviates significantly from the prior 𝑃, implying a more complex model. This complexity term acts as a regularizer. It penalizes models that are too complex and deviate significantly from the prior, thereby encouraging simpler models that are more likely to generalize well, thereby mitigating overfitting issue.
>
>
> **Q3** The organization and presentation.
>
> **A3** Thank you for your suggestions! We revised the contents in the revision.
>
> **Q4** The experimental results seem not rigorously verified. As far as I know, the CIL results on CIFAR100 dataset should start with 40-50% overall accuracy but not this low in the paper.
>
> **A4** Thank you for pointing out this!
>
> * We would like to clarify that as mentioned in lines 243-244 on page 7, the CIFAR100 dataset is divided into **10** tasks, each containing 10 classes, following the same setting as [1]. [1] reports the CIFAR100 accuracy as 37.13 for DER++ and 36.48 for ER-ACE under a memory buffer size of 500 but did not provide an error bar. Our results show DER++ at 36.37 $\pm$ 0.85 and ER-ACE at 37.05 $\pm$ 0.36, which are very similar to [1] with the same settings.
>
> * According to [2], they split CIFAR100 into **5** disjoint tasks with each task having 20 classes, reporting the accuracy of DER++ in class incremental learning as 42.08. The performance difference is **due to different task splits**: we split CIFAR100 into **10** tasks, while [2] splits it into **5** tasks. We chose the more challenging 10-task scenario over the 5-task scenario.
>
> Reference:
>
> [1] On the Effectiveness of Lipschitz-Driven Rehearsal in Continual Learning, NeurIPS 2022
>
> [2] Loss Decoupling for Task-Agnostic Continual Learning, NeurIPS 2023
>
> **Q5**  In Eq. 7, the outer minimization over is not taking any effect in the inner maximization.
>
> **A5**  Thanks for your question! We would like to clarify that the inner expectation over 𝑈 is achieved by reparameterizing the Gaussian distribution as follows: $\mu + \sigma \zeta$, where $\zeta \sim \mathcal{N}(\textbf{0}, \textbf{I})$. Therefore, the expectation for $U$ can be converted to the expectation with respect to the Gaussian distribution $\zeta$. Therefore, the outer minimization over $\theta$ still takes effect on $\theta$ after the inner max expectation. The details are illustrated as the following equation:
>
> $\max _{\mathbb{U}\in \mathcal{U}}  \mathbb{E} _{\theta \sim \mathbb{U}}  L^{CL} _{T}(\theta) \approx \mathbb{E} _{\zeta \sim \mathcal{N} (\textbf{0}, \textbf{I})}  L ^{CL} _{T}[(1 - \eta)\theta _{i}   + \eta \sigma _{n} ^2 [\nabla _{\theta} L ^{CL}(\theta _i)]+ \sigma_n \zeta]$

---

> > ### Author Response · Authors · 2024-08-11
> > **looking forward to your reply**
> >
> > Dear Reviewer YRBc,
> >
> > Thank you for your thoughtful feedback on our paper! We have done our best to address your concerns and questions. We would greatly appreciate it if you could let us know whether our response has satisfactorily addressed your concerns.
> >
> > We look forward to hearing from you.

---

> ### Comment · Reviewer_YRBc · 2024-08-11
>
> Thank you for your response. After the rebuttal, some of my concerns are solved, but not all of them. Here I would like to kindly ask for further clarification.
>
> **Q1+Q2 [Oversimplified CL Loss and Generalization Bound]** The loss function we use (for optimization and theoretical analysis) always have an underlying assumption, which is the data distribution it is evaluated. CL loss is composed of two parts, as mentioned by the authors, $\mathcal{L}\_{\text{old}}(\theta)$ and $\mathcal{L}\_{\text{new}}(\theta)$, which are typically evaluated under different data distributions: replay buffer populated from the old data distribution $\mathcal{D}\_{\text{old}}$ and the current-task dataset populated from the current data distribution $\mathcal{D}\_{\text{old}}$. **Hence it raises a serious question to the main theorem (Theorem 4.3): what is the data distribution defined $\mathcal{D}$ here?** A trivial weighted average of $\mathcal{D}\_{\text{old}}$ and $\mathcal{D}\_{\text{new}}$ is not okay as the losses are evaluated separately. Therefore I would question the validity of the theory in this paper as it to me is serious oversimplification.
>
>
> **Q4. [Low Performance of CIL models]** Are there any specific reasons for choosing these two papers as the baselines? Specifically, most of the existing work uses ResNet-32 as the backbone for CIFAR100 [1] and why do you use ResNet-18? iCaRL [2], a baseline from 7 years ago, can achieve 40+ accuracy on CIFAR100, with is conducted under the exact setting as you mentioned. The numbers in the Table are not the state-of-the-art.
>
> **Q5. [Broken MinMax Problem in Eq. 7]** Sorry I still don't get it. In the inner maximization, the expectation over $\theta$ is evaluated under the distribution $U$, it has nothing to do with the $\theta$ you have in the outer minimization. I am okay with the inner maximization being a "sensitivity penalty", can you please elaborate more on the outer $\theta$?
>
> **Additional Questions**
> - Throughout the introduction of the method (Section 3.2), there is no effective reference to existing literature. Where was the concept of "Model Sensitivity" first defined and what are existing techniques? If this concept if completely new and so as the methodology of this paper, please indicate it in the main body.
> - To me this method looks alike the idea of SAM (Sharpness-Aware Minimization) applied to CL [3]. Please discuss the relationship between this method and SAM-based CL methods.
>
> **References**
> - [1] Zhou, Da-Wei, et al. "Class-Incremental Learning: A Survey." IEEE Transactions on Pattern Analysis and Machine Intelligence (2024).
> - [2] Rebuffi, Sylvestre-Alvise, et al. "icarl: Incremental classifier and representation learning." Proceedings of the IEEE conference on Computer Vision and Pattern Recognition. 2017.
> - [3] Mehta, Sanket Vaibhav, et al. "An empirical investigation of the role of pre-training in lifelong learning." Journal of Machine Learning Research 24.214 (2023): 1-50.

---

> > ### Author Response · Authors · 2024-08-12
> > **Further Clarification (1/2)**
> >
> > Thank you for your feedback! Currently, we provide more clarifications as below.
> >
> > **Q1+Q2 [Oversimplified CL Loss and Generalization Bound]** The loss function we use (for optimization and theoretical analysis) always have an underlying assumption, which is the data distribution it is evaluated. CL loss is composed of two parts, as mentioned by the authors, $\mathcal{L}_ {\text{old}}(\theta)$ and $\mathcal{L}_ {\text{new}}(\theta)$, which are typically evaluated under different data distributions: replay buffer populated from the old data distribution $\mathcal{D}_ {\text{old}}$ and the current-task dataset populated from the current data distribution $\mathcal{D}_ {\text{new}}$. Hence it raises a serious question to the main theorem (Theorem 4.3): what is the data distribution defined $\mathcal{D}$ here? A trivial weighted average of $\mathcal{D}_ {\text{old}}$ and $\mathcal{D}_ {\text{old}}$ is not okay as the losses are evaluated separately. Therefore I would question the validity of the theory in this paper as it to me is serious oversimplification.
> >
> > **A**: $\mathcal{D}$ is defined as the CL underlying data distribution, i.e.,  $\mathcal{D}= \bigcup_{i=1}^{i=M} \mathcal{D_i}$, where $D_i$ is the data distribution for task $i$ and $M$ denotes the number of CL tasks. The generalization error **$L_{\mathcal{D}}^{CL}(\theta)$** contains the loss on unseen data, i.e., beyond $\mathcal{D}_ {\text{old}}$ and $\mathcal{D}_ {\text{new}}$, is defined as the loss on the entire CL task sequence: $L_{\mathcal{D}}^{CL}(\theta)​ = \sum_{i=1}^{M} L_ {(x, y)\sim \mathcal{D_i}} (x,y, \theta)$.  Therefore, the generalization loss we define is not an oversimplification but follows the standard definition. $\mathcal{D}_ {\text{old}}$ and $\mathcal{D}_ {\text{new}}$ are new and old task **empirical training samples** from distribution $\mathcal{D}$, respectively.
> >
> >
> >
> > **Q4. Low Performance of CIL models** Are there any specific reasons for choosing these two papers as the baselines? Specifically, most of the existing work uses ResNet-32 as the backbone for CIFAR100 [1] and why do you use ResNet-18? iCaRL [2], a baseline from 7 years ago, can achieve 40+ accuracy on CIFAR100, with is conducted under the exact setting as you mentioned. The numbers in the Table are not the state-of-the-art.
> >
> > **A**
> >
> > **why choose these two papers as baseline** We would like to clarify that we choose these two papers since they all follow the same codebase of DER++. This would enable us to conduct consistent comparisons.
> >
> > **why choose these ResNet-18** We choose ResNet-18, since this architecture is widely adopted in CL works, e.g., DER++, ER-ACE, LODE, etc.   Furthermore, our method does not depend on specific backbones. To illustrate, we integrate our method (MACL) with MEMO [2] on CIFAR100 using ResNet-32 by following the implementation in [1]. The results are shown in the following table. The dataset is split into 10 tasks, where each task has 10 classes and exemplar size is 2000. Our MACL can further improve the base method performance. We will add these discussion about the ResNet-32 results in the revision.
> >
> >
> >
> > | Method | Class-IL |
> > | -------- | -------- |
> > | MEMO    | 58.49     |
> > | MEMO + MACL    | **59.61**     |
> >
> >
> > **iCaRL can achieve 40+ accuracy on CIFAR100**. We would like to clarify that our method (MACL) is orthogonal to existing approaches and can be seamlessly integrated with various methods to enhance their performance. In our submission, we demonstrate its integration with DER++, ER-ACE, and LODE as examples, but MACL is not limited to these methods. To further illustrate its effectiveness, we conducted an additional experiment integrating MACL with iCaRL with memory size of 500. The results, shown below, demonstrate that our method outperforms iCaRL on its own.
> >
> > | Method |  Class-IL |
> > | -------- | -------- |
> > |   iCaRL   |      44.16 $\pm$  1.53   |
> > |   iCaRL + MACL   |   **48.27 $\pm$ 0.95**      |
> >
> >
> > Reference:
> >
> > [1] Class-Incremental Learning: A Survey. TPAMI 2024
> >
> > [2] A Model or 603 Exemplars: Towards Memory-Efficient Class-Incremental Learning. ICLR 2023.

---

> > > ### Author Response · Authors · 2024-08-12
> > > **Further Clarification (2/2)**
> > >
> > > This response continues the previous discussion.
> > >
> > > **Q5. Broken MinMax Problem in Eq. 7** Sorry I still don't get it. In the inner maximization, the expectation over $\theta$ is evaluated under the distribution $U$, it has nothing to do with the $\theta$ you have in the outer minimization. I am okay with the inner maximization being a "sensitivity penalty", can you please elaborate more on the outer $\theta$?
> > >
> > >
> > >
> > >
> > >
> > > **A**: Thanks for your question!   We would like to clarify that the inner expectation of the loss function can be formulated as $L(\mu, \Sigma)$, which is demonstrated in the following equations. The outer minimization is to optimize with respect to the mean of $\theta$, i.e., $\mu$. Therefore, the outer minimization takes effects on the inner maxmization. This is to make the presentation more concise. We will clarify this in the revision.
> > >
> > >
> > > The inner maximization can be converted into the following minimization optimization:
> > >
> > > \begin{align}
> > > \min_{U} [H(U) = -\mathbb{E}_{\theta \sim U(\theta)} L^{CL}(\theta) + \alpha KL(U, V)]
> > > \end{align}
> > >
> > >
> > > In our paper, we choose $\mathbb{U}(\theta) = \mathcal{N}(\theta|\mu, \Sigma)$, where $\mu$ and $\Sigma$ denote the mean and covariance, respectively. and $u(\theta)$ denotes  the density function. The above equation can be further formulated to be the following equation:
> > >
> > > \begin{align}
> > > L(\mu, \Sigma) := \mathbb{E}_{\theta \sim u(\theta)} [- L^{CL}(\theta) + \alpha [\log u(\theta) - \log v(\theta)]]
> > > \end{align}
> > >
> > > The outer minimization is with respect to $\mu$, which is the expectation of $\theta$.  This is because during inference, we only use $\mu$ as the model parameter to perform prediction.
> > >
> > >
> > > Additional Questions
> > >
> > > **Q1** Throughout the introduction of the method (Section 3.2), there is no effective reference to existing literature. Where was the concept of "Model Sensitivity" first defined and what are existing techniques? If this concept if completely new and so as the methodology of this paper, please indicate it in the main body.
> > >
> > > **A1**: Thank you for your suggestions! To our best knowledge, the concept of "Model Sensitivity" as we have defined it in our paper is indeed novel, and we have developed this methodology as an original contribution to the field. We will update the manuscript to highlight the novelty of this concept and methodology more explicitly.
> > >
> > >
> > >
> > > **Q2** To me this method looks alike the idea of SAM (Sharpness-Aware Minimization) applied to CL [3]. Please discuss the relationship between this method and SAM-based CL methods.
> > >
> > >
> > > **A2** Our method is fundamentally different from SAM-based CL [3] in several aspects.
> > >
> > >
> > > * **Deterministic vs. Probabilistic Approach**: SAM uses a fixed deterministic neighborhood, which can be restrictive in practice since updates are constrained within a fixed ball. In contrast, our method employs a probabilistic distributional approach, offering two distinct advantages: (a) The distributional neighborhood is more flexible and covers a broader range of parameter variations by sampling from a neighborhood distribution, and (b) Stochastic Gradient Descent (SGD) introduces noise during Continual Learning (CL). Our distributional approach accounts for this noise, making it a more realistic model in practice and providing stronger guarantees against parameter sensitivity.
> > >
> > > * **Uniform vs. Parameter-Specific Updates**: SAM uniformly updates all parameters, overlooking the varying importance and sensitivity of each parameter in the context of CL. Our method, on the other hand, considers these differences and treats parameters uniquely through the natural gradient with the Fisher Information Matrix (FIM). This distinction is crucial for CL, as each parameter has different sensitivity to forgetting—a factor that SAM does not address.

---

> > > ### Comment · Reviewer_YRBc · 2024-08-12
> > >
> > > Thank you for the response. I am mostly convinced by the experimental results you show in the rebuttal and I highly recommend you to report the ResNet-32 results for CIL in the revised manuscript since the current Table is too far away from the SOTA methods.
> > >
> > > As for the theory, I am still not convinced. If the data distribution is defined as $\mathcal{D}=\bigcup\_{i}\mathcal{D}\_{i}$, then the definition of the dataset $\mathcal{T} = \{(x\_i, y\_i)\}\_{i=1}^{N} \sim \mathcal{D}$ is broken, since this sampling is not reflecting what we have in CL: usually the memory and the current data samples are not evenly sampled, which is the data imbalance (for an example, please refer to UDIL [1], where they provide a generalization bound considering that). To be more specific, in Theorem 4.3,
> > > - does $\mathcal{L}\_{\mathcal{D}}^{CL}(\theta)$ in the LHS represents $\sum\_{i=1}^T\mathcal{L}\_{\mathcal{D\_i}}(\theta)$?
> > > - does $\mathcal{L}\_{\mathcal{T}}^{CL}(\theta)$ consider the data imbalance and even data absence of the previous tasks?
> > >
> > > **References**
> > > - [1] Shi, Haizhou, and Hao Wang. "A unified approach to domain incremental learning with memory: Theory and algorithm." Advances in Neural Information Processing Systems 36 (2024).

---

> > > > ### Author Response · Authors · 2024-08-13
> > > > **Further Clarification**
> > > >
> > > > Thank you for your suggestions! We greatly appreciate your feedback! We promise we will add the new results and discussion on the ResNet-32 and MEMO baselines in the revision.
> > > >
> > > >
> > > > **Q1**  does $\mathcal{L}_ {\mathcal{D}}^{CL}(\theta)$ in the LHS represents $\sum_{i=1}^T\mathcal{L}_{\mathcal{D_i}}(\theta)$?
> > > >
> > > > **A1**: Yes, it represents the $\sum_{i=1}^T\mathcal{L}_{\mathcal{D_i}}(\theta)$
> > > >
> > > > **Q2** does $\mathcal{L}_{\mathcal{T}}^{CL}(\theta)$ consider the data imbalance and even data absence of the previous tasks?
> > > >
> > > > **A2**:
> > > >
> > > > * **data imbalance**:
> > > >
> > > > **A**: In memory-replay scenarios, the dataset $T$ consists of both the new task data $\mathcal{T}_{new}$ and the memory buffer data $\mathcal{M}$. The underlying assumption in memory-replay-based approaches is that the memory buffer $\mathcal{M}$ effectively approximates the data distribution of previous tasks.
> > > >
> > > > The empirical CL loss function can be defined as:  $L^{CL}_{T}(\theta)=L _{T _{new}}(\theta)+\alpha L _{\mathcal{M}}(\theta)$,
> > > >
> > > >  where $L_{T_{new}}(\theta)$ denotes the new task empirical training loss and $L_{\mathcal{M}}(\theta)$ is the expected loss on memory buffer data. In other words, the expectation of the loss function on memory buffer, i.e., $L_{\mathcal{M}}(\theta)$, serves as an approximation of the empirical training loss from previous tasks.
> > > >
> > > >
> > > >
> > > >
> > > >
> > > > We denote the number of data examples for task $1, \cdots, T-1$ in the memory buffer $\mathcal{M}$ when training on task $T$ as $N_1, N_2, \cdots, N_{T-1}$. We also denote the number of training examples for task $T$ as $N_T$. The generalization bound, accounting for data imbalance, can be expressed as follows:
> > > >
> > > >
> > > >
> > > >
> > > > $\mathbb{E}_{\theta \sim \mathcal{N}(\mu, \Sigma)} L _ {D}^{CL}(\theta) \leq \max _ {\mathbb{U} \in \mathcal{U}} \mathbb{E} _ {\theta \sim \mathbb {U}}  L^{CL} _ {T}(\theta)  + \frac{C}{N _T + \sum _ {i=1}^{i=T-1} N_i} +     \sqrt{\frac{\tau ^2(\sqrt{q} + \sqrt{2 \log (N _ T + \sum _ {i=1} ^{i=T-1} N _ i})^2 + R + 2 \log(\frac{N _ T + \sum _ {i=1} ^{i=T-1} N _ i}{\delta})}{4(N _ T + \sum _ {i=1} ^{i=T-1} N _ {i-1})}}$
> > > >
> > > >
> > > > The data imbalance issue is reflected in the second and third terms of the generalization bound. This bound suggests that as the amount of data in the memory buffer increases (i.e., $\sum_{i=1}^{i=T-1} (N_i-1)  \uparrow$), the second and third terms in the generalization bound decrease. This is because $\lim_{x\to\infty} [h(x):=\frac{\log x}{x}] = 0$, and $h(x)$ is a decreasing function as $x$ increases over the interval $[c, \infty]$. Consequently, the generalization error decreases, which aligns with the intuitive understanding that a larger memory buffer leads to better performance.
> > > >
> > > >
> > > > * **data absense**
> > > >
> > > > **A**:   $\mathcal{L}_{\mathcal{T}}^{CL}(\theta)$  can incorporate data absense of previous tasks. For example, when the CL model learns on the task 2, the ideal task loss can be expressed as:
> > > >
> > > >    $\mathcal{L} _ {\mathcal{T}} ^{CL}(\theta) = \mathcal{L} _ {1}(\theta) + \mathcal{L} _ {2}(\theta)$, where $\mathcal{L} _ {1}(\theta)$ denotes the old task 1 empirical training loss and $\mathcal{L}_{2}(\theta)$ denotes the new task 2 empirical training loss.
> > > >
> > > >
> > > > When the data from Task 1 is unavailable while learning Task 2, we can approximate the loss function $\mathcal{L}_{1}(\theta)$ using a second-order Taylor expansion as follows:
> > > >
> > > >
> > > >   $\mathcal{L} _ {1}(\theta) \approx \mathcal{L} _ {1}(\theta_1^*) + \nabla_{\theta}\mathcal{L}_{1}(\theta_1^*)^T(\theta - \theta_1^*) +\frac{1}{2}(\theta - \theta_1^*)^TH(\theta - \theta_1^*)$
> > > >
> > > >   where $H$ is the Hessian matrix and $\theta_1^*$ is the optimal solution for task 1. Since the gradient $\nabla_ {\theta}\mathcal{L}_ {1}(\theta_1^*)$ achieves approximately optimal at $\theta_1^*$, $\nabla_{\theta}\mathcal{L}_{1}(\theta_1^*) \approx 0$. Then,
> > > >
> > > >   $\mathcal{L} _ {1}(\theta) \approx\mathcal{L}_{1}(\theta_1^*) +\frac{1}{2}(\theta - \theta_1^*)^TH(\theta - \theta_1^*)$
> > > >
> > > > The above equation is similar to EWC approach.
> > > >
> > > >   Therefore, $\mathcal{L}_{\mathcal{T}}^{CL}(\theta)$ can incorporate the case that the data from previous tasks are unavailable.

---

> > > > > ### Comment · Reviewer_YRBc · 2024-08-13
> > > > >
> > > > > Thank you for your response.
> > > > >
> > > > > I'm now happy with the data imbalance case, and it shares certain level of the similarity of the bound presented in the previous work [1]. I think this result should be included in the revision to make the original theorem more specific and not an overly generalized one.
> > > > >
> > > > > I am still curious about the case of data absence:
> > > > > - how is it reflected in the generalization bound or does it affect the generalization bound at all?
> > > > > - a more general case would be considering imbalance and absence at the same time, encompassing both of them in one generalization bound.
> > > > >
> > > > > I have raised my rating to 5 accordingly. I think this submission can be a significant contribution to the community in the future if a set of improved empirical results and more accurate theorems stated. I would sincerely thank the authors for responding to my questions and I look forward to an improved manuscript in the near future (regardless of whether it is accepted).
> > > > >
> > > > > **References**
> > > > > - [1] Shi, Haizhou, and Hao Wang. "A unified approach to domain incremental learning with memory: Theory and algorithm." Advances in Neural Information Processing Systems 36 (2024).

---

> > > > > > ### Author Response · Authors · 2024-08-14
> > > > > > **Further Clarification**
> > > > > >
> > > > > > Thank you very much for your response and support! We sincerely appreciate your valuable suggestions! We will certainly revise our paper according to your comments.
> > > > > >
> > > > > >
> > > > > > **Q1** I think this result should be included in the revision to make the original theorem more specific and not an overly generalized one.
> > > > > >
> > > > > > **A1** Thank you for your suggestions! We will ensure the improved theorem be included in the revision.
> > > > > >
> > > > > > **Q2** data absence generalization bound: how is it reflected in the generalization bound or does it affect the generalization bound at all?
> > > > > >
> > > > > > **A2** We denote the number of training data examples for task $1, \cdots, T-1$  when training on task $T$ as $N_1, N_2, \cdots, N_{T-1}$. We also denote the number of training examples for the new task $T$ as $N_T$. The generalization bound, considering data absence, can be formulated as follows:
> > > > > >
> > > > > > $L _ {T}^{CL}(\theta) \approx L _ {T_{new}}(\theta) + \lambda L _ {T_{old}^{appr}}(\theta)$
> > > > > >
> > > > > > $\mathbb{E}_{\theta \sim \mathcal{N}(\mu, \Sigma)} L _ {D}^{CL}(\theta) \leq \max _ {\mathbb{U} \in \mathcal{U}} \mathbb{E} _ {\theta \sim \mathbb {U}}  L^{CL} _ {T}(\theta)  + \frac{C}{N _T + \lambda \sum _ {i=1}^{i=T-1} N_i} +     \sqrt{\frac{\tau ^2(\sqrt{q} + \sqrt{2 \log (N _ T + \lambda \sum _ {i=1} ^{i=T-1} N _ i})^2 + R + 2 \log(\frac{N _ T + \lambda \sum _ {i=1} ^{i=T-1} N _ i}{\delta})}{4(N _ T + \lambda \sum _ {i=1} ^{i=T-1} N _ {i})}}$
> > > > > >
> > > > > >  We denote $\lambda \sum _ {i=1} ^{i=T-1} N _ {i}$ as the effective sample size from previous tasks since the loss function on the training data from previous tasks is approximated due to the absence of data from earlier tasks. The parameter $\lambda$ reflects the trade-off between learning the new task and retaining knowledge from previous tasks. A larger $\lambda$ imposes stronger regularization on the first term on the RHS of the generalization bound, ensuring that the parameters learned for the new task do not deviate significantly from those learned on previous tasks. Consequently, the first term increases, indicating that the overall empirical training loss rises. Meanwhile, the second and third terms on the RHS decrease. The third term, representing the simplified KL divergence between the posterior and prior, decreases, indicating that the posterior distribution for the new task remains close to the parameter distribution from previous tasks. This occurs because we prioritize preserving knowledge from earlier tasks, which in turn constrains the learning process for the new task.
> > > > > >
> > > > > > **Q3** a more general case would be considering imbalance and absence at the same time, encompassing both of them in one generalization bound.
> > > > > >
> > > > > > **A3**  Yes, we can have a more general theory bound as below:
> > > > > >
> > > > > > $\mathbb{E}_{\theta \sim \mathcal{N}(\mu, \Sigma)} L _ {D}^{CL}(\theta) \leq \max _ {\mathbb{U} \in \mathcal{U}} \mathbb{E} _ {\theta \sim \mathbb {U}}  L^{CL} _ {T}(\theta)  + \frac{C}{N _T + \lambda \sum _ {i=1}^{i=T-1} N_i} +     \sqrt{\frac{\tau ^2(\sqrt{q} + \sqrt{2 \log (N _ T + \lambda \sum _ {i=1} ^{i=T-1} N _ i})^2 + R + 2 \log(\frac{N _ T + \lambda \sum _ {i=1} ^{i=T-1} N _ i}{\delta})}{4(N _ T + \lambda \sum _ {i=1} ^{i=T-1} N _ {i})}}$
> > > > > >
> > > > > > when $\lambda = 1$, and $L^{CL}_{T}(\theta)=L _{T _{new}}(\theta)+\alpha L _{\mathcal{M}}(\theta)$, the effective sample size from previous tasks is equal to the memory buffer size. The generalization bound corresponds to memory replay CL approach.
> > > > > >
> > > > > > when $\lambda = \beta$, and $L _ {T}^{CL}(\theta) \approx L _ {T_{new}}(\theta) + \lambda L _ {T_{old}^{appr}}(\theta)$, the bound correspond to regularization-based approach, the effective sample size from previous tasks is equal to $\lambda-$ weighted sample size from previous tasks since the loss function  on the training data from previous tasks is approximated due to the absence of data from earlier tasks. The generalization bound corresponds to data absense of CL approach.

---

### Official Review · Reviewer_1NyD · 2024-07-16

**Soundness:** 3
**Presentation:** 3
**Contribution:** 3
**Rating:** 7
**Confidence:** 4

**Summary:**

The paper proposes a new perspective on stability-plasticity trade-off in continual learning, centered on controlling model sensitivity to model updates. The goal is then to ensure that alteration in model parameters does not negatively impact the CL performance of the model.

In order to solve this challenging task, the authors propose to optimise model’s performance based on the worst-case scenario of parameter distributions within a distribution neighborhood. Given the intractability of this problem due to the infinite dimension of the space of all possible distribution within this neighborhood. The authors solve this issue by modelling the model parameters by a Gaussian distribution. They then propose to solve a min-max problem, with a CL objective equal to the sum of a cross entropy loss and  a forgetting mitigation loss (which can come from a replay method, a regularization term, a gradient projection loss, ...) within a neighborhood based on the KL-divergence with the current CL parameter distribution. In order to efficiently solve this optimization problem:
* The authors leverage natural gradient descent in the distribution parameter space.
* They develop an mirror-descent method in the dual space alleviating the need for computing the Fisher information matrix.

The authors support their method with a theoretical analysis and some experiments. The theoretical analysis shows a reduction in loss variance, indicating a lower model parameter sensitivity, and tighter generalization bounds. The experiments were conducted using a ResNet18 backbone, Split CIFAR and Tiny-ImageNet benchmarks, and adding the proposed approach to a large number of CL algorithms. They also include ablation study to analyse the hyperparameters, study the effect of memory size, comparing natural gradient descent to SGD and analysing the method efficiency.

**Strengths:**

* The approach takes a novel and innovative (to the best of my knowledge) perspective on solving the stability-plasticity trade-off.

* The method is based on solid theoretical development. It theoretically leads to the intended effect, and experimental results confirm this theoretical insights.

* The method is easily applicable on top of existing CL algorithms, for what seems to be a limited computational cost. It also seems to be applicable in many CL scenarios.

* The method is tested with a large number of CL algorithms, showing interesting boost in performance. I also appreciate the ablation study which provides interesting insights.

* The authors provide a detailed description of the implementation, which increases the reproducibility of the work.

**Weaknesses:**

The main weakness of the paper is the lack of diversity in the evaluation settings, and this in multiple aspects:

* The evaluation is limited to benchmarks that lack in diversity in the data distribution. I invite the authors to look at other benchmarks (e.g. 5-datasets (used in different recent works, e.g. Learning to Prompt for Continual Learning, Wang et al. 2022), Nevis'22 (Bornschein et al. 2023), ...

* The evaluation is limited to relatively short sequences, it would be interesting to analyse the effect of the sequence length.

* The evaluation is limited to the task aware scenario, although it is a priori also suitable for the more challenging Online Continual Learning setting.

* The evaluation is limited to a single architecture that is trained from scratch. It would be interesting to analyse the impact of architecture, and of initialization on the method.

The authors also analyse the efficiency of the method in one setting only. It would be interesting to add other methods, as the additional cost can be a fix one, and the small additive cost in the analyzed setting can be due to the high cost of the original CL algorithm.

Despite these limits, I think the work already provides interesting perspective and insights.

**Questions:**

* Is modelling the current CL parameters with a standard normal distribution realistic? Isn't it too restrictive? What if we start the CL process with an already pretrained model?

* Can this modelling assumption be changed with a more general Gaussian distribution? What would be the impact on the derivation?

* Can the method be added to the recent family of learning to prompt techniques for CL?

**Limitations:**

The work is of foundational nature. I think discussing potential negative societal impact is out of scope.

---

> ### Author Rebuttal · Authors · 2024-08-06
>
> We extend our sincere appreciation for your constructive feedback!
>
> **Q1** 5-datasets evaluation
>
> **A1** Thank you for your suggestions! We perform experiments on 5-datasets (memory buffer size of 500) with our method (MACL) as below.
>
> | Method | Class-IL | Task-IL |
> | -------- | -------- | -------- |
> | ER     |  66.03 $\pm$ 1.37    |    92.58 $\pm$ 1.26 |
> | ER+MACL     |  **67.32 $\pm$ 1.18**    |    **93.21 $\pm$ 1.08** |
> | DER++     | 85.92  $\pm$ 0.33    |   87.16  $\pm$ 0.21    |
> | DER++MACL    | **87.23  $\pm$ 0.51**    |   **87.51  $\pm$ 0.30**    |
>
> **Q2** analyse the effect of the sequence length or long sequence.
>
> **A2**  We invite you to refer to the **global response Q2 and A2**.
>
> **Q3**  Online Continual Learning setting evaluation.
>
> **A3**  Thank you for your suggestions! We invite you to refer to the **global response Q1 and A1**.
>
> **Q4**  analyse the impact of architecture, and of initialization on the method.
>
> **A4**  Thank you for your question! We conducted an additional experiment using a pre-trained Vision Transformer (ViT), specifically the vit-base-patch16-224 model pre-trained on ImageNet1K, following existing CL literature settings. The results, shown in the following table for CIFAR100 with DER++ and a memory size of 500, demonstrate that using a pre-trained ViT significantly improves CL performance. In addition, integrating MACL with DER++ further enhances the CL performance with the pre-trained ViT.
>
> | architecture | ResNet | ViT |
> | -------- | -------- | -------- |
> | Task-IL    |  75.64 $\pm$ 0.60   |   96.72 $\pm$ 0.31   |
> | Task-IL + MACL    |   **77.53 $\pm$ 0.89**             |   **97.31 $\pm$ 0.46**   |
> | Class-IL   | 36.37 $\pm$ 0.85     |   76.21 $\pm$ 0.67   |
> | Class-IL + MACL  | **39.42 $\pm$ 0.82**        |   **77.83 $\pm$ 0.80**              |
>
> **Q5**  analyse the efficiency of other CL methods
>
> **A5** Thank you for pointing this out! We conducted experiments to compare the running efficiency by integrating our proposed method with multiple CL approaches, including DER++, ER-ACE, and LODE. The training time (in seconds) for a single epoch on CIFAR100 is shown below. Our method increases the training time by approximately 55% to 61%.
>
> **running time on CIFAR100 (seconds)**
>
> | Method | w/o MACL | w/ MACL |
> | ------ | -------- | ------ |
> | DER++  | 8.7      | 13.5   |
> | ER-ACE | 6.3      | 10.2  |
> | LODE   | 13.2     | 20.8   |
>
> **Q6** Is modelling the current CL parameters with a standard normal distribution realistic? Isn't it too restrictive? What if we start the CL process with an already pretrained model?
>
> **A6** Thank you for your question! We acknowledge that this assumption might be restrictive in certain scenarios. When starting the CL process with an already pretrained model, the parameters may not be initially distributed according to a standard normal distribution. In such cases, our method can be adapted to account for the existing parameter distribution of the pretrained model. We initialize our approach using the pretrained model's parameter statistics to estimate a more general Gaussian distribution, ensuring a more accurate and realistic modeling process.  Furthermore, we invite you to the **Q7** and **A7** for a more general Gaussian distribution derivation. By doing so, our method retains its effectiveness while being flexible enough to accommodate different starting conditions, whether the model begins from scratch or with pretrained parameters. This adaptability helps maintain the generalizability and flexibility of our approach across various settings.
>
> **Q7** change assumption with a more general Gaussian distribution?
>
> **A7** Thank you for your suggestion! Here, we assume a more general Gaussian distribution to model current CL parameter distribution, $V(\theta) = \mathcal{N}(\theta|\mu_0, \Sigma_0)$, where $\mu_0$ and $\Sigma_0$ denote the mean and covariance matrix of current CL model parameters, respectively.  We can obtain the following update equation:
>
>
> $\Sigma^{-1}_{i+1}= (1 - \eta)\Sigma^{-1}_i + \eta \mathbb{E} _{\theta\sim u(\theta)} [-\nabla _{\theta\theta}^2 \mathcal{L} ^{CL}(\theta) + \Sigma_0^{-1}]$
>
> $\mu_{i+1} = (1 - \eta)\mu_{i} + \eta \Sigma _{i+1} \mathbb{E} _{\theta \sim u(\theta)} [\nabla _{\theta} \mathcal{L}^{CL}(\theta)+ \mu_0 \Sigma _0^{-1}]$
>
> Compared to standard normal distribution modeling assumption. Here, the above more general Gaussian parameter distribution incorporates one additional term of $\mu_0 \Sigma_0^{-1}$ and $\Sigma_0^{-1}$ for $\mu$ and $\Sigma$ update process, respectively. This allows for greater flexibility and adaptability in the update process.
>
> **Q8** add to learning to prompt techniques for CL?
>
> **A8**  Thank you for your suggestions! We conducted an experiment integrating our proposed method (MACL) with the SOTA prompt-based CL method, CODA-Prompt [1]. Our method operates on the parameters of prompt components and corresponding keys/attention vectors. The results on ImageNet-R are shown in the following table. The improvement is due to reducing the sensitivity of these parameters. This way, when the model learns new tasks, parameter changes do not significantly increase the loss. During inference, the weighted prompt remains more stable, mitigating forgetting. Additionally, our method reduces overfitting to new tasks, thus improving overall performance.
>
> **CODA Prompt Results on ImageNet-R**
> | number of tasks | 10 | 20 |
> | -------- | -------- | -------- |
> | CODA-P    |  75.45 $\pm$ 0.56   |  72.37 $\pm$ 1.19   |
> | CODA-P + MACL    | **76.39 $\pm$ 0.67**     | **73.42 $\pm$ 1.23**    |
>
> Reference:
>
> [1] CODA-Prompt: COntinual Decomposed Attention-based Prompting for Rehearsal-Free Continual Learning, CVPR 2023

---

> > ### Comment · Reviewer_1NyD · 2024-08-13
> > **Answer to rebuttal**
> >
> > I appreciate the extensive answers of the authors to all the reviews, and I find that many of the added results strengthen the paper. I thank the authors for their efforts, and raise my score accordingly.

---

> > > ### Author Response · Authors · 2024-08-13
> > > **Thank you!**
> > >
> > > Thank you for your updates! We deeply appreciate your support!

---

> ### Comment · Area_Chair_yQnh · 2024-08-12
> **Reviewer-author discussion period will end soon.**
>
> Dear reviewer, authors have provided their rebuttal. Can you please check it and provide your response? Reviewer-author discussion period will end very soon. Thanks.
> - AC

---

### Author Rebuttal · Authors · 2024-08-06

# Global Response


**Q1**   Online Continual Learning setting evaluation.

**A1**  Thank you for your suggestions! We conduct experiment in the online continual learning setting on the SOTA approach,  MKD with PCR [1,2]. We follow the same dataset split and hyperparameter setting as [1]. The results by integrating our method (MACL) with MKD on CIFAR100 and Tiny-ImageNet are shown in the following tables. MACL further improves the online CL performance.


**Online CL Results on CIFAR100 under the blurry boundary setting**
| Memory Size | 1000  | 2000  | 5000|
| ----------- | ---- | ---- | --- |
| MKD(PCR)         | 35.6 $\pm$ 0.66     | 44.95 $\pm$ 0.42    |  54.87 $\pm$ 0.39   |
| MKD(PCR) + MACL       | **37.2 $\pm$ 0.53**       |  **46.17 $\pm$ 0.51**   |**56.21 $\pm$ 0.43**     |

**Online CL Results on Tiny-ImageNet under the blurry boundary setting**
| Memory Size | 2000  | 5000  | 10000|
| ----------- | ---- | ---- | --- |
| MKD(PCR)         |  17.33 $\pm$ 1.28    |  29.58 $\pm$ 0.6   |  38.02 $\pm$ 1.64   |
| MKD(PCR) + MACL       | **18.21 $\pm$ 1.32**     |  **30.69 $\pm$ 0.71**    | **38.73 $\pm$ 1.56**    |


**Q2** analyse the effect of the sequence length or long sequence.

**A2** We conducted experiments by splitting **Tiny-ImageNet** (200 classes) into sequences of different lengths, specifically 10 and 20 tasks. The results for task-incremental learning (Task-IL) and class-incremental learning (Class-IL) using DER++ with a memory size of 500 are shown in the following table. These results indicate that even with longer task sequences, our method (MACL) still provides improvements over the compared methods, i.e., **more than 3.3\% in Task-IL on a sequence of 20 tasks**.

| number of tasks | 10   | 20   |
| --------------- | ---- | ---- |
| Class-IL        |$19.38\pm1.41$      |   15.02 $\pm$ 0.53   |
| Class-IL + MACL |**20.17 $\pm$ 1.56**    |   **16.08 $\pm$ 0.81**   |
| Task-IL         | $51.91\pm0.68$ |51.65 $\pm$ 1.36  |
| Task-IL  + MACL |**54.03 $\pm$ 0.79**  |**54.96 $\pm$ 0.72** |


**Q3** Apply the proposed method (MACL) for learning to prompt techniques for CL?

**A3**  Thank you for your suggestions! We conducted an experiment integrating our proposed method (MACL) with the SOTA prompt-based CL method, CODA-Prompt [3]. MACL operates on the parameters of prompt components and corresponding keys/attention vectors. The results on ImageNet-R are shown in the following table. The improvement is due to reducing the sensitivity of these parameters. This way, when the model learns new tasks, parameter changes do not significantly increase the loss. During inference, the weighted prompt remains more stable, mitigating forgetting. Additionally, MACL reduces overfitting to new tasks, thus improving overall performance.

**CODA Prompt Results on ImageNet-R**
| number of tasks | 10 | 20 |
| -------- | -------- | -------- |
| CODA-P    |  75.45 $\pm$ 0.56   |  72.37 $\pm$ 1.19   |
| CODA-P + MACL    | **76.39 $\pm$ 0.67**     | **73.42 $\pm$ 1.23**    |


**Q4** More state-of-the-art (SOTA) baseline.

**A4** We added one SOTA class-incremental learning (CIL) MRFA [4], compared the performance on ImageNet100 with memory size of 2000. "10-10" means there are 10 classes in the base task, and each following incremental task also has 10 classes. "50-10" means there are 50 classes in the base task, and each incremental task has 10 classes. The results are shown below.

| Method | 10-10 | 50-10 |
| -------- | -------- | -------- |
| MRFA(iCaRL)    |   67.34    |   65.15    |
| MRFA(iCaRL) +MACL   | **67.82**       | **65.76**     |

Reference

[1] Rethinking Momentum Knowledge Distillation in Online Continual Learning, ICML 2024

[2] PCR: Proxy-based Contrastive Replay for Online Class-Incremental Continual Learning, CVPR 2023

[3] CODA-Prompt: COntinual Decomposed Attention-based Prompting for Rehearsal-Free Continual Learning, CVPR 2023

[4] Multi-layer Rehearsal Feature Augmentation for Class-Incremental Learning, ICML 2024

---

### Decision · Program_Chairs · 2024-09-25

**Decision:**

Accept (poster)

**Comment:**

This work considers model sensitivity for improving continual learning. An optimization approach based on parameter distribution is introduced to reduce forgetting and overfitting. While there seems to be a room for improvement, the authors provided sufficient updates to address the concerns regarding empirical evidence. Therefore, an acceptance is recommended.

For the camera-ready submission, please make sure to improve the manuscript as much as possible to reflect the reviewers' feedback and concerns. In particular, please make sure to have the data distribution $D$ well defined and to have the generalization bound addressed sufficiently comprehensively. Updated empirical results should be reflected at least in the appendix with proper comments in the main text.

Even though not commented by the reviewers, it might be also helpful to address the connection of this work to the existing flat-minima/SWAD related works.